# An unusual case of childhood osteoarticular tuberculosis from the Árpádian Age cemetery of Győrszentiván-Révhegyi tag (Győr-Moson-Sopron county, Hungary)

**Olga Spekker**[1]*, **Luca Kis**[1], **Andrea Deák**[2], **Eszter Makai**[3], **György Pálfi**[1], **Orsolya Anna Váradi**[1,4], **Erika Molnár**[1]

**1** Department of Biological Anthropology, University of Szeged, Szeged, Hungary, **2** Rómer Flóris Museum of Art and History, Győr, Hungary, **3** Department of Radiology, University of Szeged, Szeged, Hungary, **4** Department of Microbiology, University of Szeged, Szeged, Hungary

* olga.spekker@gmail.com

**Data Availability Statement:** All relevant data are within the manuscript and its Supporting Information files.

## Abstract

Ancient human remains exhibiting bony changes consistent with osteoarticular tuberculosis (OATB) indicate that the disease has afflicted mankind for millennia. Nonetheless, not many pediatric OATB cases have been published in the paleopathological literature–from Hungary, only three cases have been described up to now. In our paper, we demonstrate a child (**S0603**) from the Árpádian Age cemetery of Győrszentiván-Révhegyi tag (northwestern Hungary), who represents a unique case of OATB regarding both the pattern and severity of the observed bony changes. During the macromorphological and radiological investigations, the most serious alterations were discovered in the upper thoracic spine–the development of osteolytic lesions led to severe bone loss and consequent collapse and fusion of several adjacent vertebrae. The pathological process terminated in a sharp, rigid angular kyphosis. Disruption of the normal spine curvature resulted in consequent deformation of the whole thoracic wall–it became "rugby-ball-shaped". The overall nature and pattern of the detected alterations, as well as their resemblance to those of described in previously published archaeological and modern cases from the pre-antibiotic era indicate that they are most consistent with OATB. Based on the severity and extent of the lesions, as well as on the evidence of secondary healing, **S0603** suffered from TB for a long time prior to death. Besides body deformation, OATB resulted in consequent disability in daily activities, which would have required regular and significant care from others to survive. It implies that in the Árpádian Age community of Győrszentiván-Révhegyi tag, there was a willingness to care for people in need. Detailed archaeological case studies can give us a unique insight into the natural history and different presentations of OATB. Furthermore, they can provide paleopathologists with a stronger basis for diagnosing TB and consequently, with a more sensitive means of assessing TB frequency in past populations.

**Funding:** This work was funded by the University of Szeged Open Access Fund (5006) to OS. The National Research, Development and Innovation Office (Hungary) (K 125561) and the "Árpád-ház Program" (39509/2018/KFSZ) of the Hungarian Ministry of Human Capacities provided funding for GP. The funders had no role in study design, data collection and analysis, decision to publish, or preparation of the manuscript.

**Competing interests:** The authors have declared that no competing interests exist.

## Introduction

Despite growing interest in the last few decades, pediatric tuberculosis (TB) as a main cause of morbidity and mortality, especially in TB-endemic regions, remains relatively neglected today [1–6]. However, particular attention should be paid to eliminate or at least control TB in the foreseeable future [1, 2, 5–7]. Childhood TB is an important TB control indicator, since it reflects recent and/or ongoing transmission in the community, as it is usually acquired postnatally from an adult contact with clinically active TB disease [2, 3, 8–10]. Children and adolescents with latent TB infection represent reservoirs for future transmission following disease reactivation, often many years after the primary infection; and therefore, can provide the source of future epidemics [2, 3, 8]. According to the latest Global Tuberculosis Report from the World Health Organization (WHO), children under the age of 15 years accounted for approximately 11% of all TB cases and 13.8% of all TB deaths in 2018 –an estimated 1.1 million children became ill with TB and about 205,000 children died of TB worldwide [7]. Nevertheless, the actual global burden of pediatric TB is very likely higher as substantial challenges in diagnosis and surveillance compromise the quality of epidemiological data on the disease [1, 3, 4, 7, 9–11].

Once infected with TB bacteria (i.e., members of the *Mycobacterium tuberculosis* complex; MTBC), children are at particularly higher risk of rapid progression to clinically active TB disease than adults, with the vast majority of them (>95%) developing it within the first 12 months after exposure [1, 6, 8–13]. Moreover, children are more prone to develop severe, extra-pulmonary forms of TB (e.g., tuberculous meningitis and miliary TB) that are associated with high morbidity and mortality [1, 2, 6, 8, 13–15]. Age-related differences in both the innate and adaptive immune responses to TB may play a crucial role in increasing the vulnerability of children to the disease compared to adults [1, 13]. Based on pre-chemotherapy literature data from the first half of the 20th century, Marais and his co-workers [12] identified two high-risk periods of childhood for progression to clinically active TB disease following primary infection: infancy (less than 2 years of age) and adolescence (more than 10 years of age).

Although in most pediatric cases (up to 80%), TB presents as pulmonary disease, other parts of the human body, including the skeletal system, can also be affected [6, 15, 16]. Osteoarticular or skeletal TB (i.e., tuberculous involvement of the bones and/or joints; OATB) is more frequent in children than in adults, accounting for approximately 10–35% of pediatric extra-pulmonary TB cases and about 5–7% of all pediatric TB cases [17–22]. OATB usually arises secondary to hematogenous seeding of TB bacteria from an often unknown primary site of infection outside the skeleton into the bone and/or synovial tissue during or shortly after the mycobacteremic phase of primary infection or late reactivation of the disease [21, 23, 24]. Less commonly, lymphogenous dissemination, contiguous spread from adjacent structures or direct inoculation of TB bacteria into a skeletal site can also occur [21, 25, 26]. Virtually any bone or joint of the human body can be affected by the disease–the three main forms of OATB are spinal TB (i.e., combination of tuberculous vertebral osteomyelitis and arthritis; ∼50%), tuberculous osteomyelitis of the extra-spinal bones (∼11%), and tuberculous arthritis of the extra-spinal joints (∼30%) [18, 19, 23, 27]. Although the aforementioned forms of OATB are usually present alone, their concomitant occurrence can be observed in some cases [23]. Its relative rarity and highly variable clinical and radiological presentations make pediatric OATB a diagnostic challenge in the modern medical practice [28, 29]. However, early diagnosis is crucial to improve the clinical outcome, as OATB can be a debilitating medical condition with serious and potentially irreversible orthopedic and/or neurologic complications, even many years after the onset of the disease [18, 19, 30, 31].

Recent evolutionary genetics studies on the age of the MTBC by Comas and his co-workers [32] revealed that the human-adapted members of the complex may have co-evolved with

their host to successfully infect, cause disease, and transmit over tens of thousands of years [33–36]. Besides genetic findings, ancient human skeletons and mummies exhibiting bony changes consistent with OATB [e.g., 37–41] also indicate that the disease has afflicted mankind for millennia [34, 35, 42]. Nonetheless, not many cases with OATB in children (less than 15 years of age) have been published in the paleopathological literature [e.g., 43–50]. The afore-mentioned studies reporting archaeological and modern pediatric OATB cases from the pre-antibiotic era have pointed out that the wide variety of manifestations of the disease observed in patients today were present in prehistoric and historic communities [49]. Besides meticu-lous descriptions from the literature from the first half of the 20th century (pre-chemotherapy era), detailed archaeological case studies also provide a unique insight into the natural history and different presentations of OATB in children. They help us in identifying non-pathogno-monic bony changes and/or patterns of lesions that can later be used as diagnostic criteria for OATB in both the modern medical and paleopathological practices. By scrutinizing these new diagnostic criteria in living patients with similar alterations, a more appropriate diagnosis could be established that may contribute to improving the clinical outcome of the disease in those patients. It should be noted that the bony changes observed in archaeological cases may differ from those detectable in living patients, due in part to the introduction of antibiotics in the management of TB from the second half of the 20th century [51]. However, in both devel-oping and developed countries, with the emergence of multidrug-resistant TB (in which the applied antibiotic therapy is often not effective enough), presentations of OATB that are simi-lar to those of discovered in archaeological cases from the pre-antibiotic era may become more common in the future. On the other hand, by providing physicians and paleopathologists with a stronger basis for diagnosing different manifestations of OATB, a more sensitive means of assessing the disease frequency in current and archaeological populations can be achieved.

In our paper, we demonstrate a child (i.e., **S0603**) from the Árpádian Age cemetery of Győrszentiván-Révhegyi tag (Győr-Moson-Sopron county, Hungary), who represents a unique case of pediatric OATB with multifocal involvement of the axial skeleton (i.e., spine and ribs) regarding both the pattern and severity of the observed bony changes, as well as the archaeological context.

## Materials and methods

Between 2014 and 2015, prior to the construction works of main road #813 bypassing the city of Győr (Győr-Moson-Sopron county, northwestern Hungary), test and preventive excava-tions were carried out under the direction of Andrea Deák at the merged archaeological sites of Győr–Győrszentiván and Győr–Révhegyi tag, geographically located in close vicinity to the present-day city of Győrszentiván (Fig 1A). Remains of settlements from the Bronze Age, Iron Age, Roman period, and Árpádian Age, as well as of cemeteries from the Roman period and Árpádian Age were also discovered in the large excavation area that coincided with the trail of main road #813.

From the Árpádian Age cemetery of the archaeological site (Fig 1B), a total of 57 burials were unearthed that, based on the grave goods, can clearly be dated to the 11th–12th centuries CE. It should be noted that the Árpádian Age cemetery is only partially excavated, and its exact extent is not known. The 57 burials can be divided into two main groups. The earlier (11th c. CE), western grave group consisted of 11 burials covering a semicircular area of the cemetery, whereas the later (12th c. CE), eastern grave group, which was contiguous with the eastern end of the first one, was comprised of rows of burials. In the uniformly orientated, rect-angular pit-graves, the individuals were buried in an extended supine position with the head at the southwestern end of the grave–consistent with the west-east grave orientation typical of

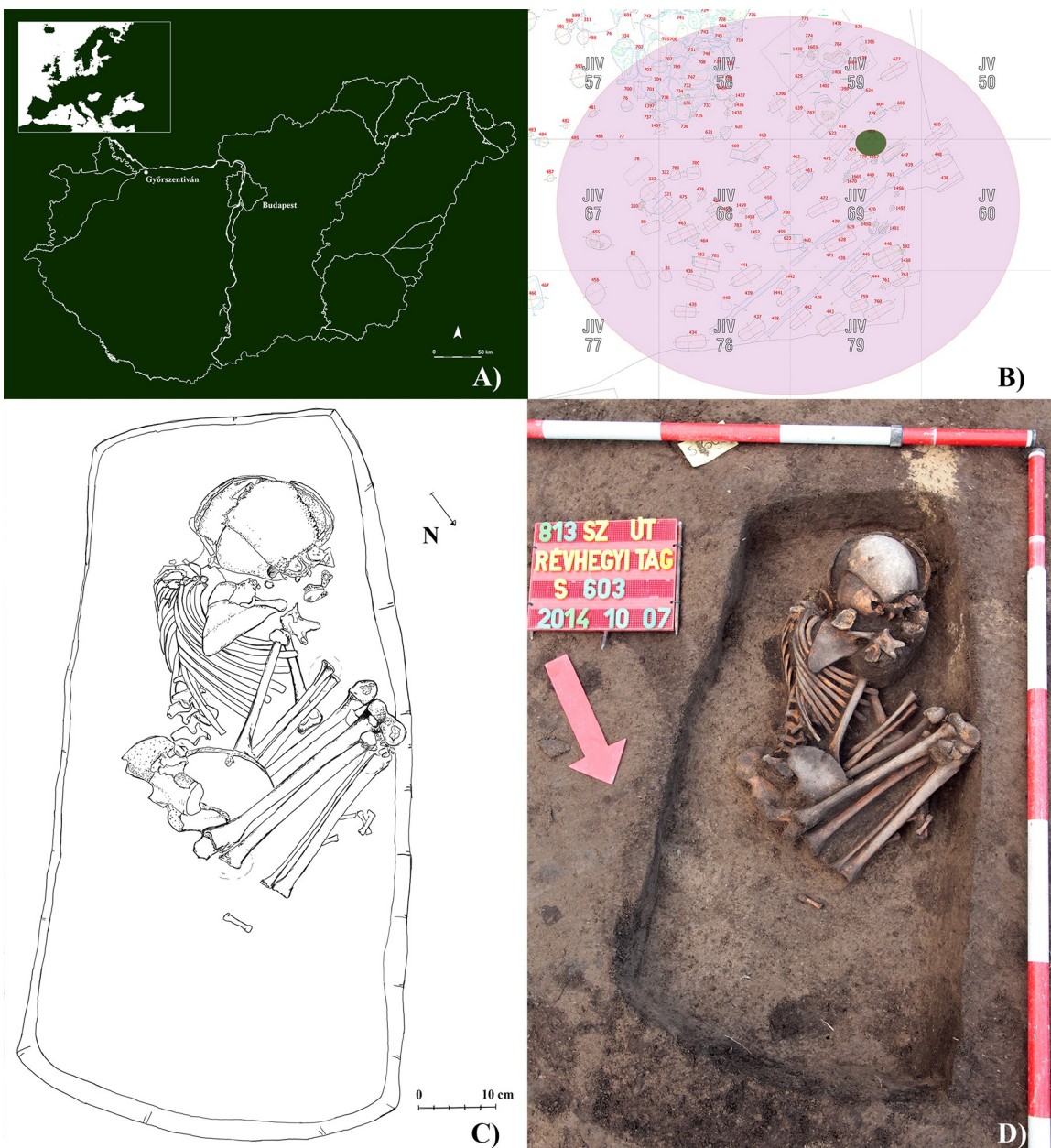

**Fig 1.** A) Map of Hungary showing the location of the Győrszentiván-Révhegyi tag archaeological site; B) Plan drawing of the Árpádian Age cemetery of Győrszentiván-Révhegyi tag archaeological site with the location of burial S0603; C) Drawing of the burial of S0603 *in situ*; and D) Photo of the burial of S0603 *in situ*.

Árpádian Age burials. Although the grave of **S0603** was in one of the 12^th^-century-CE grave rows of the cemetery, the body of the deceased was buried in a flexed position, with the opposite orientation to that of the other individuals from the cemetery and without any grave goods (Fig 1B–1D).

Along with the remains of the 56 other individuals, the skeleton of **S0603** is temporarily housed at the Department of Biological Anthropology, University of Szeged (Szeged, Hungary). Based on tooth formation and development [52], tooth eruption pattern [53], and diaphyseal length of long tubular bones [54], **S0603** was around 12 years of age at the time of

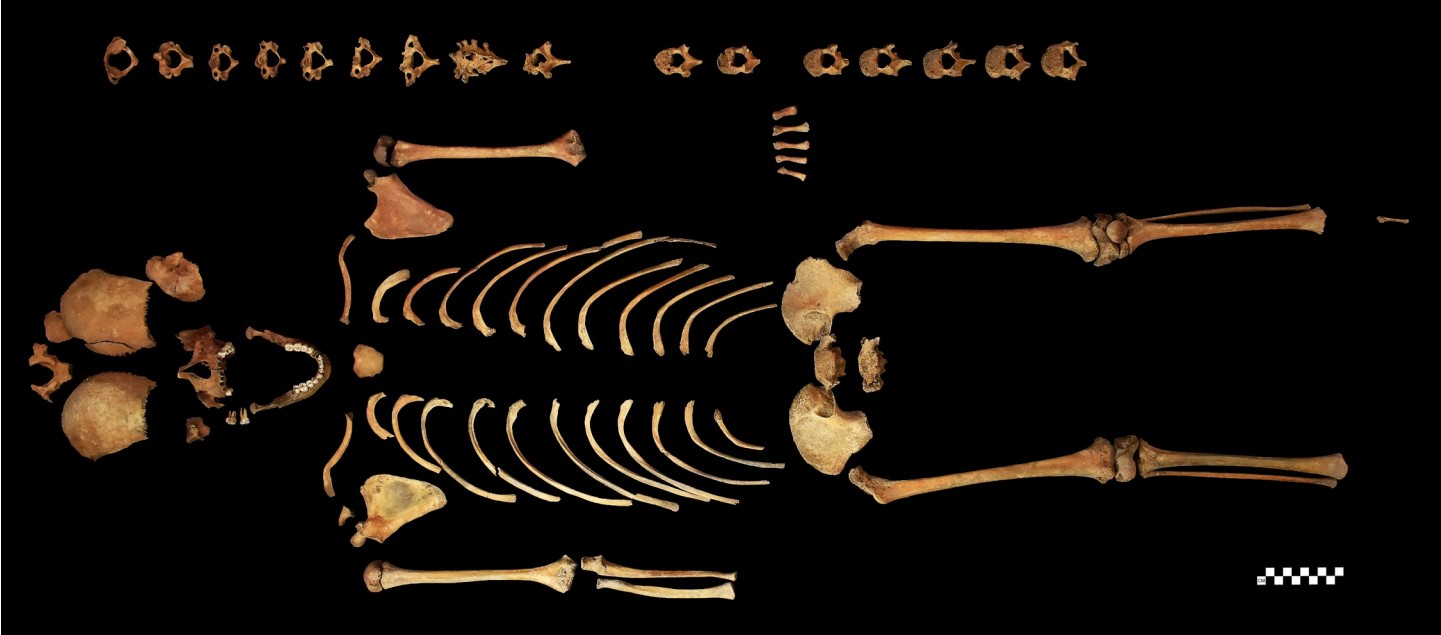

**Fig 2. The skeleton of S0603.**

death. Both the qualitative and quantitative state of preservation of the child's skeletal remains are quite good (Fig 2). The skeleton was subject to a detailed macromorphological investigation, focusing on the detection of pathological bony changes. CT imaging of the T1–6 region of the spine was also performed to improve the diagnosis established on the basis of prior macroscopic observations–it was carried out with a Philips Brilliance iCT 256.

## Ethics statement

Specimen number: **S0603**.

The skeleton evaluated in the described study is housed in the Department of Biological Anthropology, University of Szeged, in Szeged, Hungary. Access to the specimen was granted by the Department of Biological Anthropology, University of Szeged (Szeged, Hungary) and the Rómer Flóris Museum of Art and History (Győr, Hungary).

No permits were required for the described study, complying with all relevant regulations.

## Results

In the 12-year-old child's skeletal remains, a number of severe pathological bony changes were macroscopically observed, predominantly in the axial skeleton (i.e., spine and ribs).

## Skull

In the skull, both parietal bones exhibited signs of *cribra cranii* in close vicinity to the joining point of the sagittal and lambdoid sutures (Fig 3). On the ectocranial surface of the occipital bone, pitting and slight cortical remodeling were observed on its basal and lateral parts; the pathological process spared the articular surfaces (Fig 4). Similar pitting was detected on the inferior surface of the left temporal pyramid, adjacent to the affected occipital areas (Fig 4).

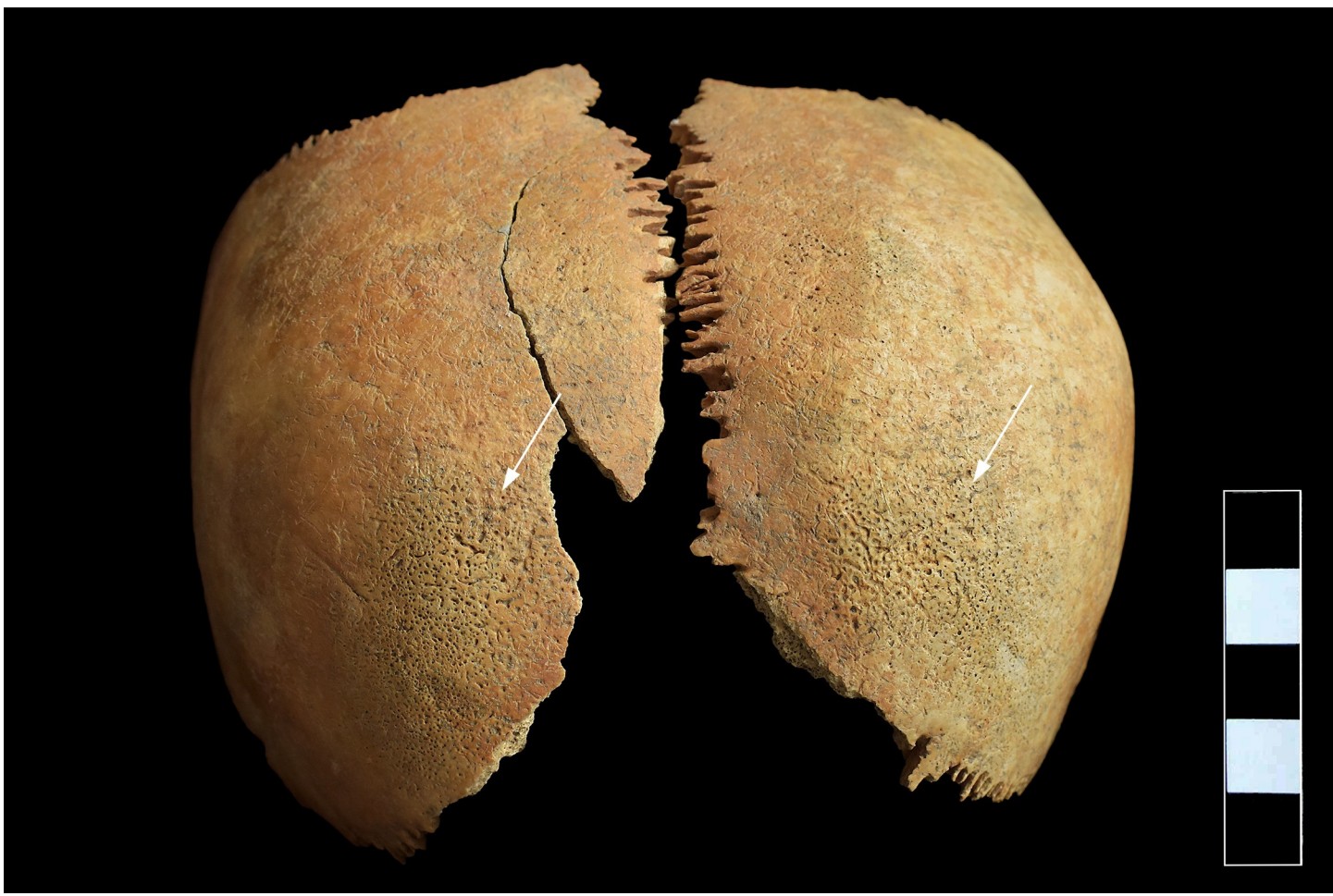

**Fig 3. Signs of *cribra cranii* on the ectocranial surface of both parietal bones (white arrows).**

## Spine

In the vertebral column, the C1–5 vertebral bodies showed signs of inflammation in form of pitting and slight cortical remodeling, particularly on their anterior aspect and costal processes (Fig 5). The original height of the C1–5 vertebral bodies was maintained (Fig 5). In the C6–7 vertebral bodies, the pathological process affected not only the cortical bone layers but also the trabeculae: multiple, well-circumscribed osteolytic lesions corresponding to round, oval or slightly lobulated necrotizing foci were detected (Fig 6A–6D and 6F). The presence of osteolytic lesions led to considerable bone loss, predominantly in the antero-inferior parts of the C6–7 vertebral bodies; because of the severe bone destruction, they became wedge-shaped (Fig 6E). In the C6–7 vertebral bodies, the pathological process extended towards the transverse processes: pitting and slight cortical remodeling were noted on both sides (Fig 6A, 6C and 6F).

The most severe pathological bony changes were observed in the upper thoracic spine: the development of multiple, well-circumscribed osteolytic lesions resulted in almost complete destruction of the T1–5 vertebral bodies with disappearance of the intervening disc spaces and formation of a severe angular kyphosis in the cervicothoracic region of the spine (Fig 7 and S1 Video). Subsequently, the small, wedge-shaped remnants of the T1–5 vertebral bodies fused

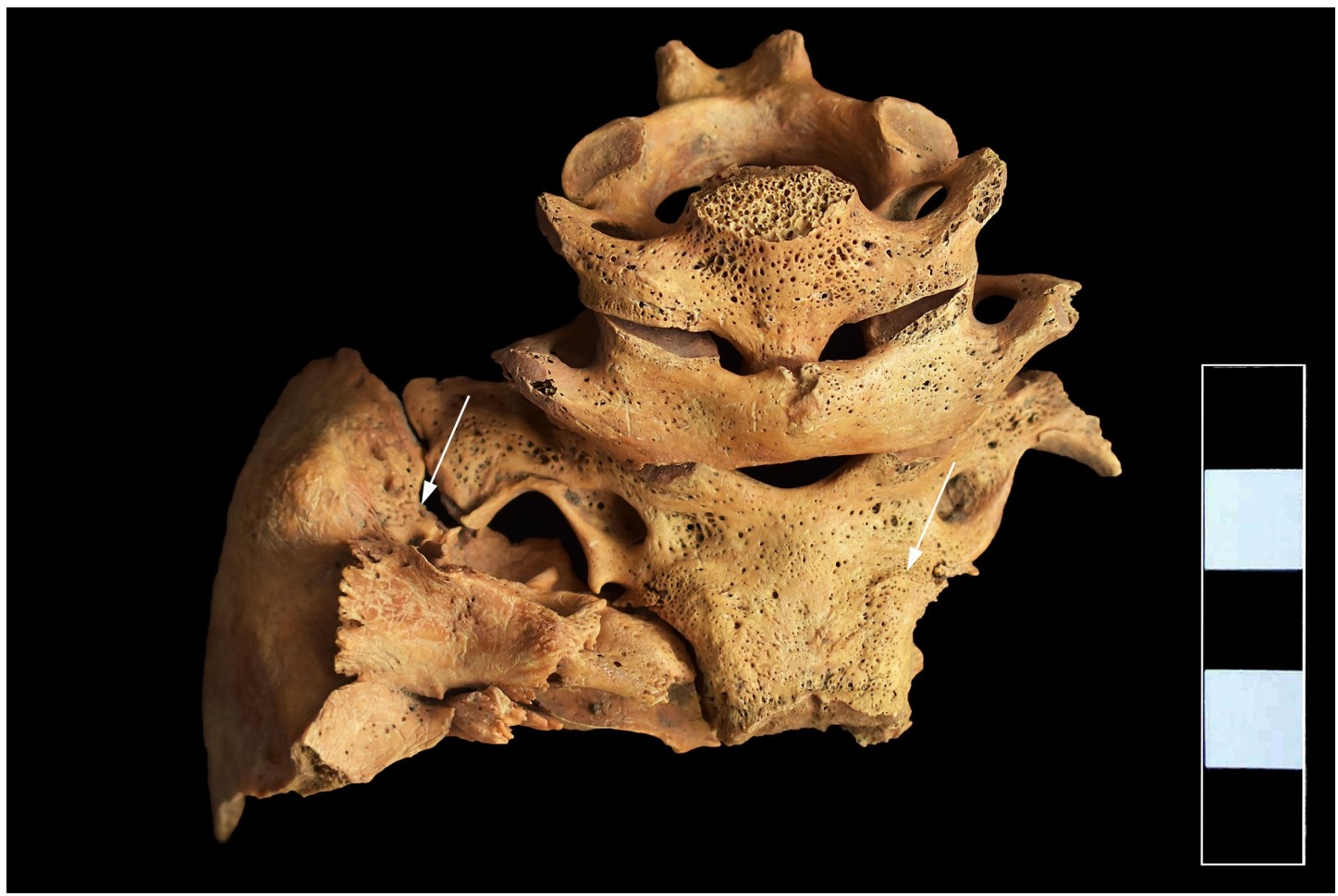

**Fig 4. Pitting and slight cortical remodeling on the ectocranial surface of the occipital and left temporal bones (white arrows).**

together. The anterior part of the T6–7 bodies was almost completely destroyed by the pathological process; however, the T5–6 and T6–7 intervening disc spaces and the body height of T6 and T7 were maintained (Fig 7A, 7C and 7D). The posterior part of the T6–7 vertebral bodies was also affected: abnormal porosity, multiple, variable-sized, well-circumscribed osteolytic lesions with thin sclerotic margins, pitting, and slight cortical remodeling were detected in their remnants (Fig 7A, 7C and 7D). The upper thoracic vertebral arci and their processes also showed signs of inflammation in form of pitting, slight cortical remodeling, and osteolytic lesions, especially in the T4–7 region (Fig 7A–7C and 7E). Ankylosis of the intervertebral joints and fusion of the laminae were also noted in the upper thoracic spine (T1–6) (Fig 7B, 7E and S1 Video). Furthermore, the CT imaging of the T1–6 region revealed that the vertebral canal was slightly narrowed and altered in shape, especially at the apex of the severe angular kyphosis.

In the lower thoracic (Fig 8A and 8B) and lumbar spine (Fig 9A and 9B), multiple, well-circumscribed osteolytic lesions, cortical remodeling, and signs of hypervascularization were recorded on the anterior and lateral aspects of all the preserved vertebral bodies. Moreover, they presented swelling. The lower thoracic and lumbar vertebral arci and their processes also exhibited pitting and slight remodeling of the cortical bone layers (Fig 8C).

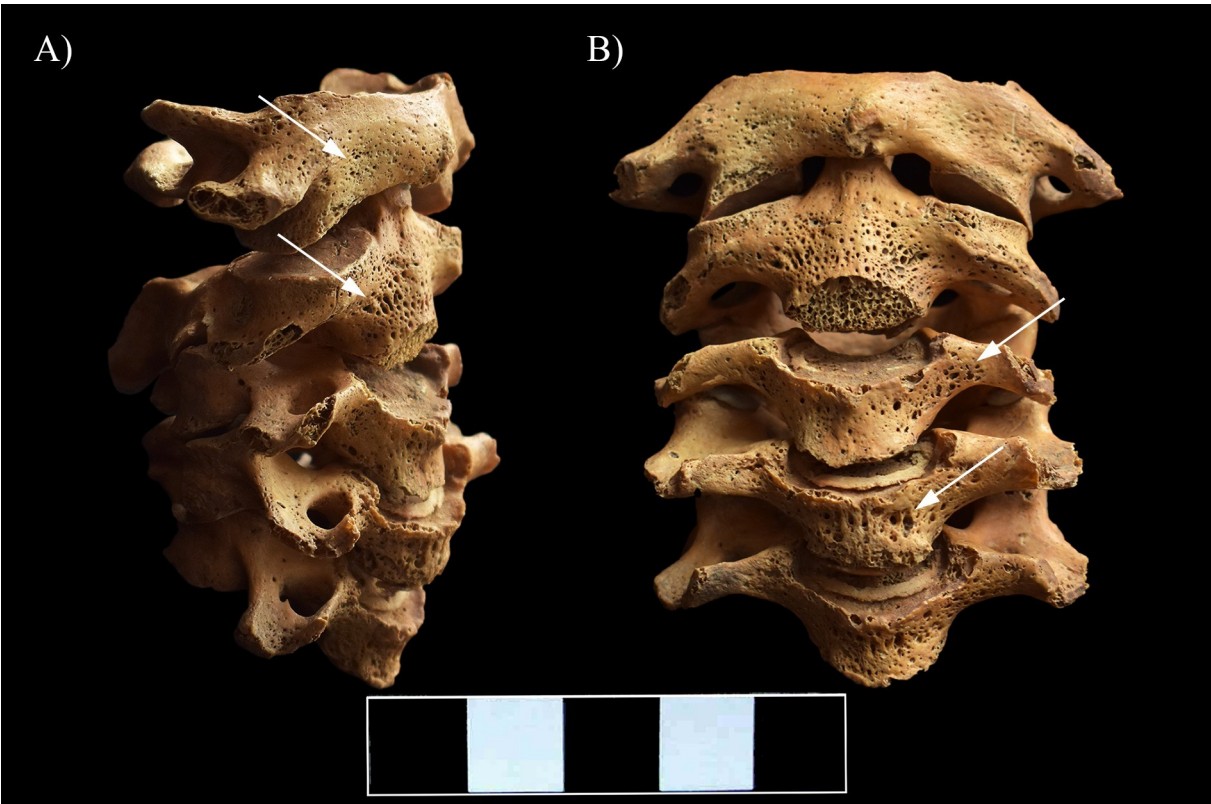

**Fig 5. Pitting and slight cortical remodeling on the C1–5 vertebrae (white arrows).** A) Lateral view (right side) and B) Anterior view.

### Ribs

On both sides of the thoracic wall, almost all the preserved ribs showed pathological bony changes, predominantly on their vertebral end adjacent to the affected thoracic vertebral bodies. The vertebral ends of the first six ribs, especially from the 3rd to the 6th, were supero-inferiorly flattened and thinned, and displayed an abnormal curvature (Fig 10A). An irregular morphology of the articular area was also observed from the 4th to the 6th ribs (especially on the left side) (Fig 10A). From the 7th to the 11th ribs, the vertebral ends presented swelling and pitting (the 12th ribs are missing *post-mortem*) (Fig 10B). Furthermore, they were supero-inferiorly flattened and their original curvature changed (Fig 10B).

### Clavicles

Although the clavicles exhibited no signs of inflammation, similar to the vertebral ends of the ribs, they showed an abnormal curvature on both the sternal and acromial ends (Fig 11).

### Pelvic area

On the right iliac bone, reactive new bone formation covering the whole iliac fossa of the iliac wing was detected (Fig 12). The proximal end of the right femur displayed similar alterations: periosteal new bone formation was noted on its anterior, medial, and posterior surfaces until the level of the lesser trochanter (Fig 13).

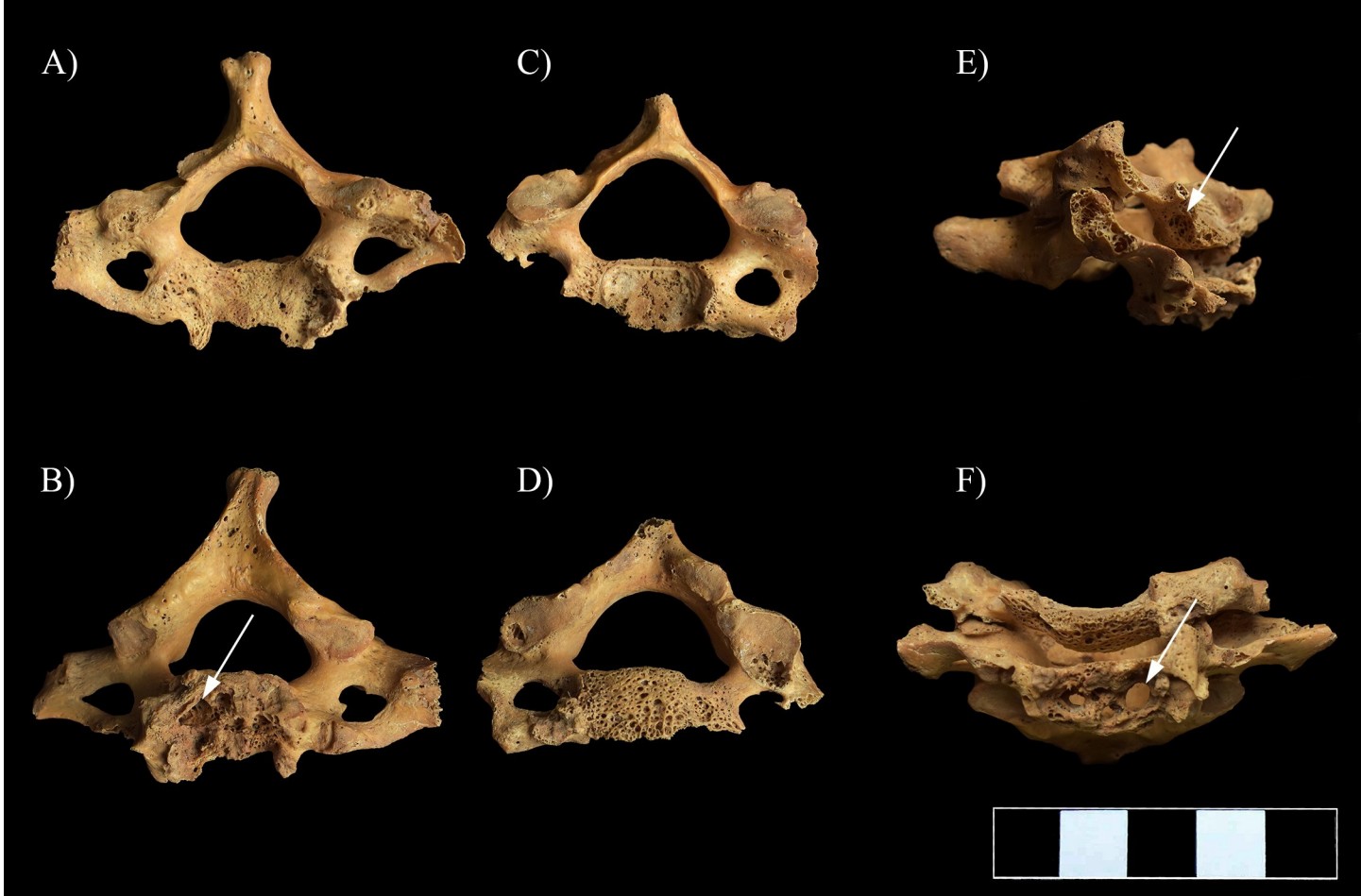

**Fig 6. Multiple, well-circumscribed osteolytic lesions (white arrows) and consequent severe bone loss in the C6–7 vertebrae.** A) Superior view of C7, B) Inferior view of C7, C) Superior view of C6, D) Inferior view of C6, E) Lateral view of C6–7 (right side), and F) Anterior view of C6–7.

## Discussion

The numerous pathological alterations (mainly of a lytic nature with very little bone formation), which were observed in the skeletal remains of **S0603**, indicate that the 12-year-old child suffered from a systemic infectious disease. It affected various regions of the skeleton but the most remarkable bony changes were discovered predominantly in the spine–in form of well-circumscribed osteolytic lesions leading to severe bone loss and collapse and fusion of several adjacent vertebrae. The pathological process terminated in a localized sagittal spinal deformity with a sharp angulation (i.e., sharp, rigid angular kyphosis of the upper thoracic spine). Disruption of the normal curvature of the vertebral column resulted in consequent deformation of the whole thoracic wall: besides the spinal changes, the altered shape of the clavicles and ribs (especially of their vertebral ends) gave the thoracic wall a "rugby-ball-shaped" appearance. The signs of inflammation on the vertebral ends of ribs, as well as of an overlying paravertebral cold abscess in the pelvic area (i.e., new bone formation on the iliac fossa of the right iliac bone and on the proximal end of the right femur) are suggestive of direct extension of the infection from the vertebral column into the adjacent ribs and soft tissues. The overall nature and pattern of the detected bony changes, as well as their resemblance to those of described in

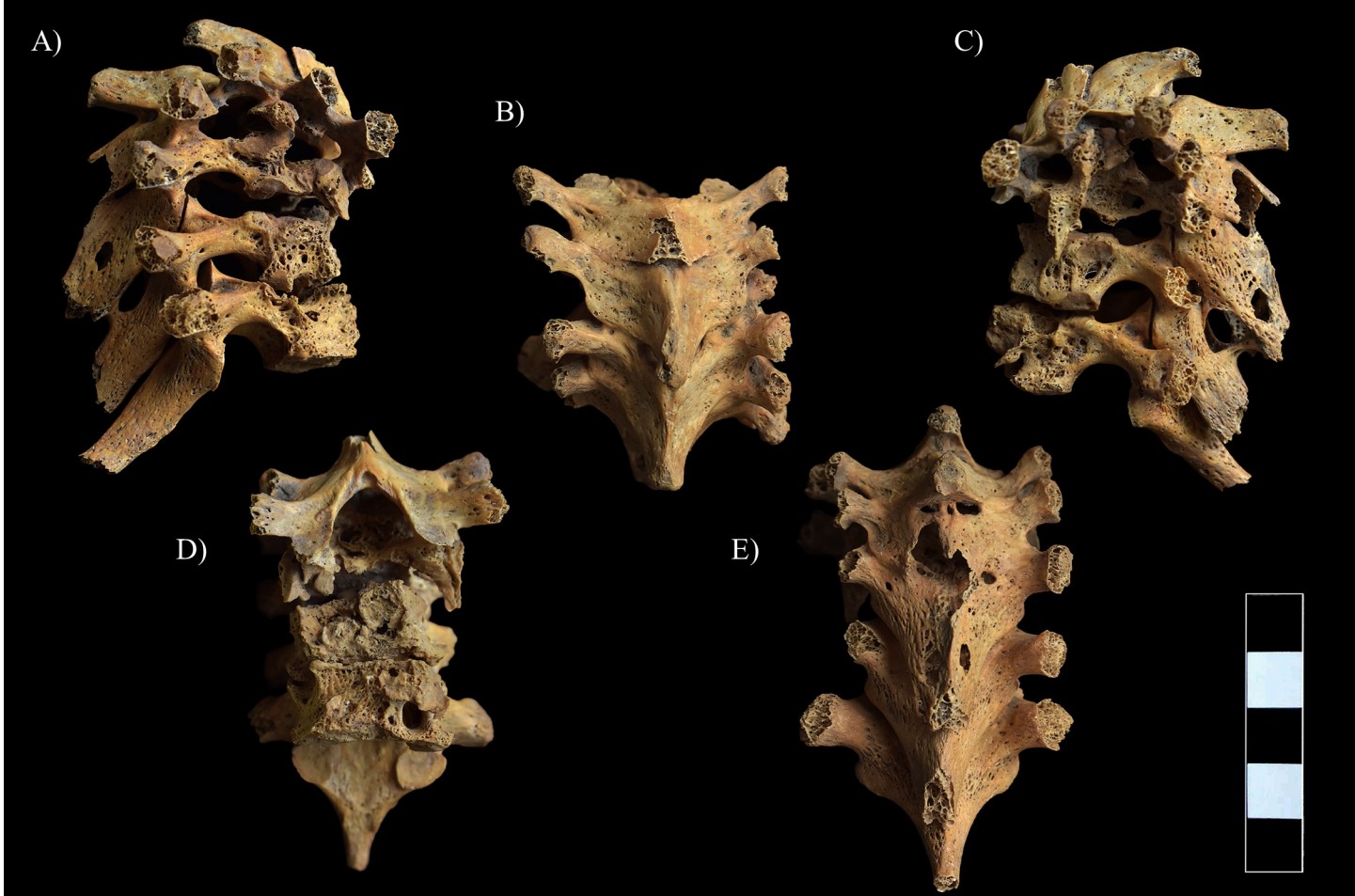

**Fig 7. Pott's gibbus in the T1–7 region.** A) Lateral view (right side), B) Superior view, C) Lateral view (left side), D) Anterior view, and E) Posterior view.

previously published archaeological and modern cases from the pre-antibiotic era [e.g., 45, 49, 50, 55, 56] indicate that they are most consistent with pediatric OATB.

In the vast majority of the cases (90–95%), spinal TB–also known as TB spondylitis or Pott's disease–arises from the anterior subchondral (paradiscal) region of the vertebral body [19, 21, 23, 57–59]. This area has a dense end-arteriolar network that makes it susceptible to bacterial seeding via the segmental arterial circulation [21, 23, 57, 59, 60]. Lodgment of TB bacteria into the anterior subchondral region triggers the onset of granulomatous inflammation of the cancellous bone with tubercle formation in the red bone marrow [21, 23, 61]. The development and caseous necrosis of the gradually expanding and coalescing tubercles induce the growth of the initial intra-vertebral abscess and the establishment of additional intra-vertebral abscesses within the affected vertebral body [23, 61, 62]. Furthermore, TB involvement of the segmental artery branches terminating in the anterior subchondral region generates deprivation of the blood supply to the cancellous bone [62]. Any or all of the aforementioned processes result in necrosis and consequent resorption of the bone trabeculae that ultimately lead to the formation of osteolytic lesions in the anterior subchondral region, with subsequent involvement of the entire vertebral body and occasionally of the posterior vertebral elements [23, 30, 57, 59, 60, 62]. As the pathological process progresses, not only the trabeculae but also the cortical bone layers of the affected vertebral body can become destroyed; and thus, the

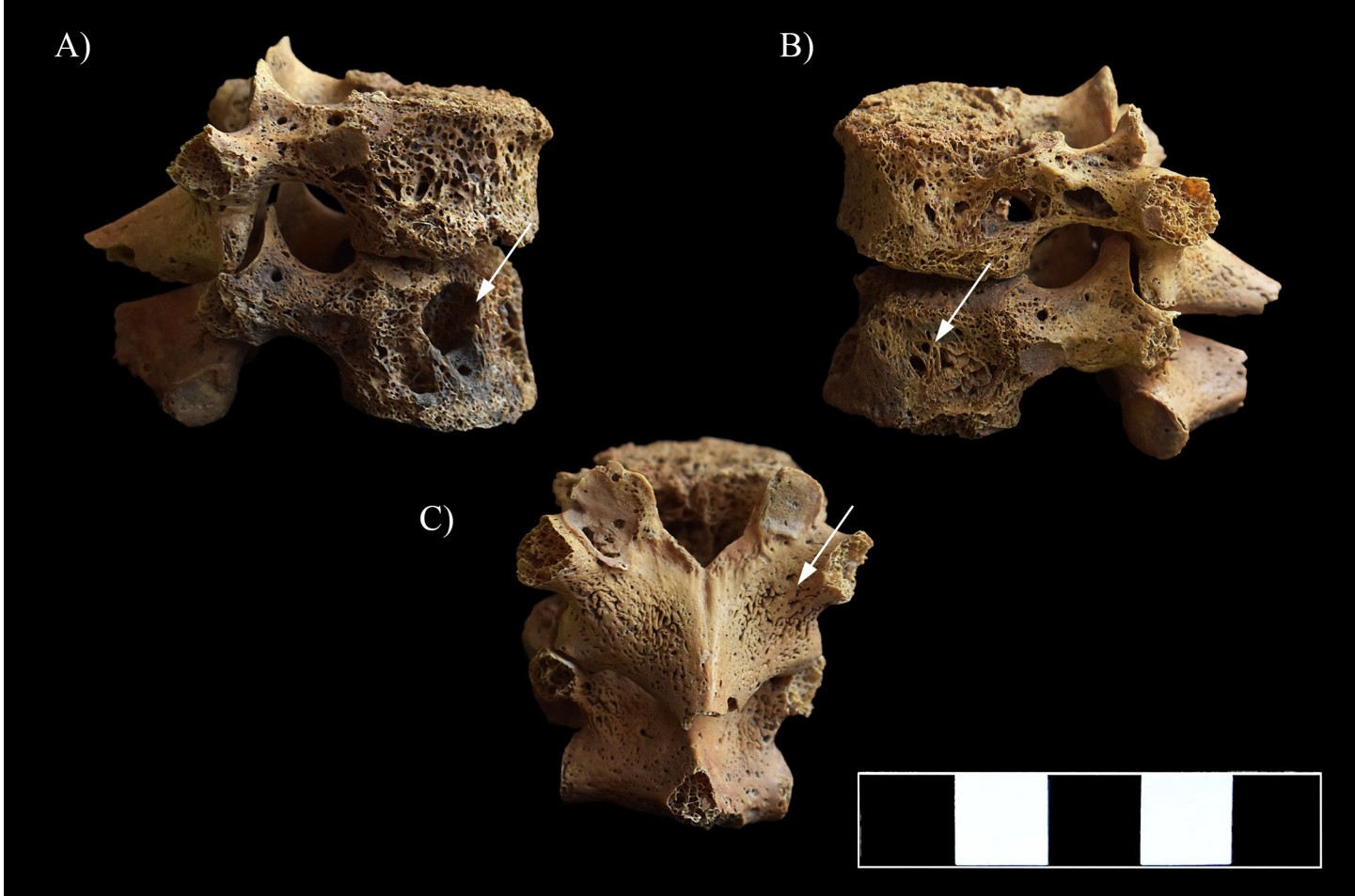

**Fig 8. Multiple, well-circumscribed osteolytic lesions, cortical remodeling, and signs of hypervascularization in the T11–12 vertebrae (white arrows).** A) Lateral view (right side), B) Lateral view (left side), and C) Posterior view.

intra-vertebral abscess can extend towards the sub-ligamentous space, the adjoining intervertebral discs or the adjacent soft tissues (e.g., ligaments and muscles) [23, 30, 57, 59, 60].

The anterior longitudinal ligament (i.e., a fibrous structure that covers the anterior and lateral surfaces of the vertebral bodies), at least temporarily, resists the horizontal progression of the infection [62]. Thus, the TB mass extending into the sub-ligamentous space (i.e., an extra-vertebral abscess) can spread only vertically (upwards or downwards) beneath the ligament from the initially affected vertebral body into a similar location at one or more contiguous vertebrae or beyond [62]. The sub-ligamentous extension of the extra-vertebral abscess results in stripping of the periosteum and anterior longitudinal ligament from the anterior and lateral vertebral surfaces [62]. This generates deprivation of the periosteal blood supply to the affected vertebral bodies with consequent ischemia [28, 61, 62]. The combination of pressure and ischemic effects caused by the presence and spread of the extra-vertebral abscess in the sub-ligamentous space results in shallow cortical erosion on the anterior and lateral vertebral surfaces that gives them a scalloped appearance (i.e., anterior gouge defect) [23, 28, 62]. Besides cortical erosion, reactive new bone formation can also occur on the vertebral surfaces underlying the extra-vertebral abscess [23, 63]. As the pathological process progresses, it can involve not only

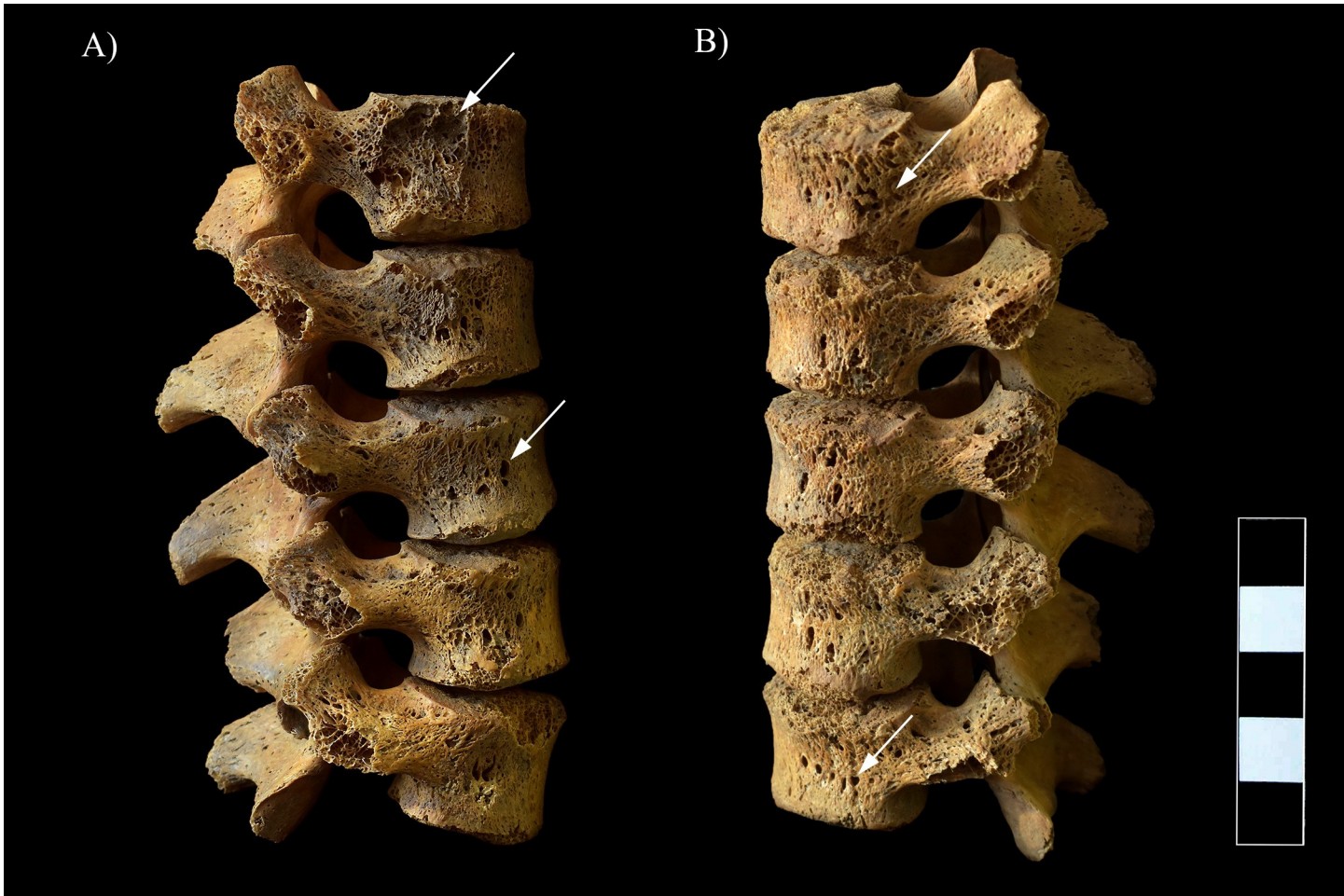

**Fig 9. Multiple, well-circumscribed osteolytic lesions, cortical remodeling, and signs of hypervascularization in the L1–5 vertebrae (white arrows).** A) Lateral view (right side) and B) Lateral view (left side).

the cortical bone layers of the affected vertebral bodies but also their trabeculae, since avascular vertebrae are more susceptible to infection [28, 63].

Until more advanced stages of spinal TB, the intervertebral disc is relatively spared by the infection–most likely due to the lack of proteolytic enzymes in TB bacteria [18, 19, 23, 58, 59, 61]. However, the progressive destruction of the subchondral region of two adjacent vertebrae can compromise the nutrition of the adjoining intervertebral disc [57]. This can lead to disc degeneration and/or result in disc herniation into the weakened adjoining vertebral bodies, with gradual diminution or eventual loss of the intervening disc space [64]. Degenerated and/ or herniated discs are more prone to seeding by TB bacteria from the subchondral cancellous bone either via sub-ligamentous dissemination or contiguous spread; and therefore, to be secondarily involved by the infection [57]. In children, not only the vertebral bodies but also the intervertebral discs can represent the initial site of infection due to their vascularized nature [59, 61, 65].

Extension of the intra-vertebral abscess into the adjacent soft tissues can result in the formation of an extra-vertebral cold abscess (i.e., a slowly progressive abscess without characteristic signs of inflammation (such as heat, erythema or tenderness), which can become encapsulated and calcified over time) [59, 60, 63, 66, 67]. In spinal TB, the development of an extra-vertebral

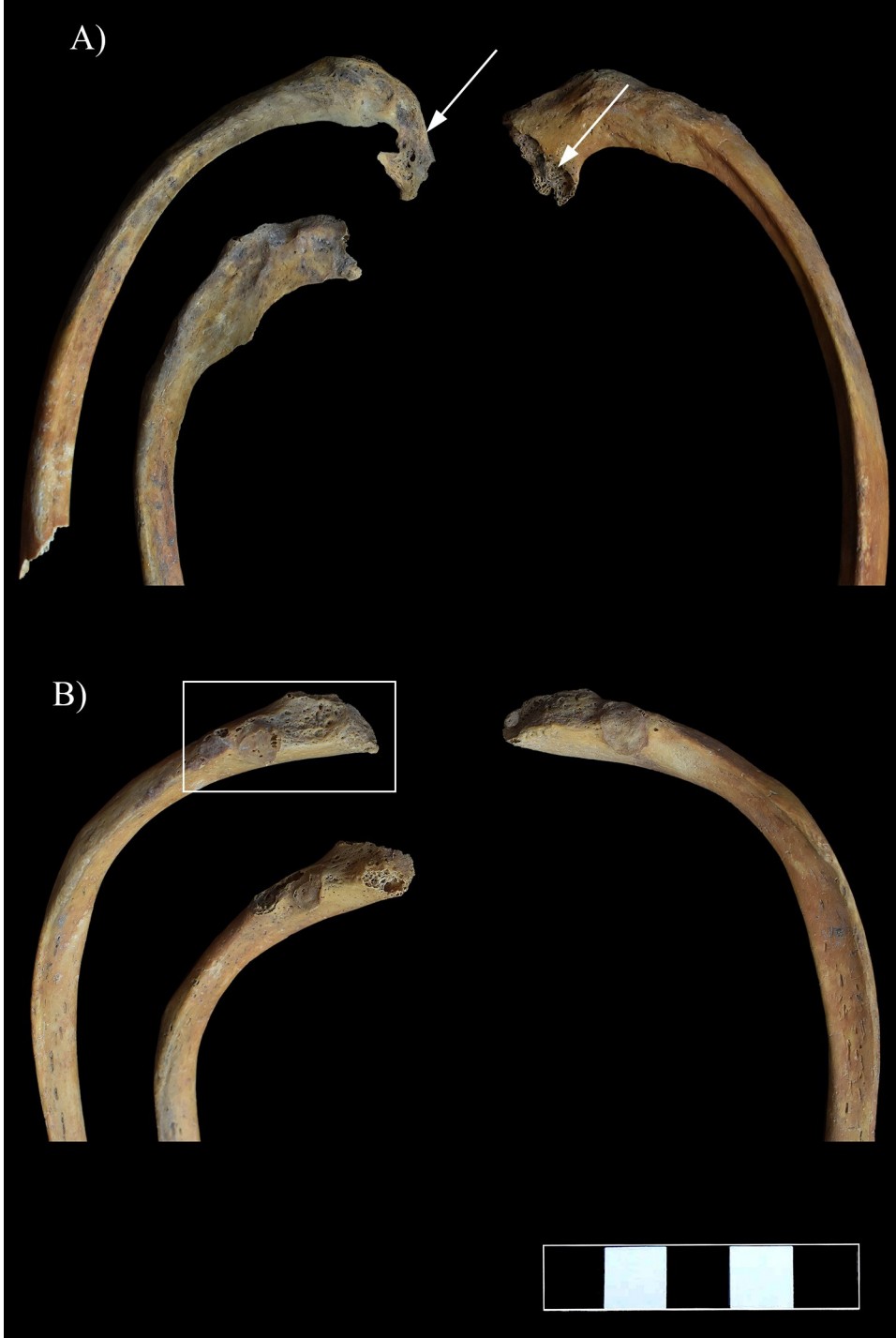

**Fig 10.** A) Supero-inferior flattening, thinning, and destructive articular changes (white arrows) and B) swelling and pitting (white rectangle) on the vertebral end of some ribs.

cold abscess is a common complication–it occurs in about two-thirds of the cases [60, 66]. The TB mass can accumulate within the prevertebral and/or paravertebral spaces with the formation of prevertebral or paravertebral cold abscesses that are commonly associated with fistulae

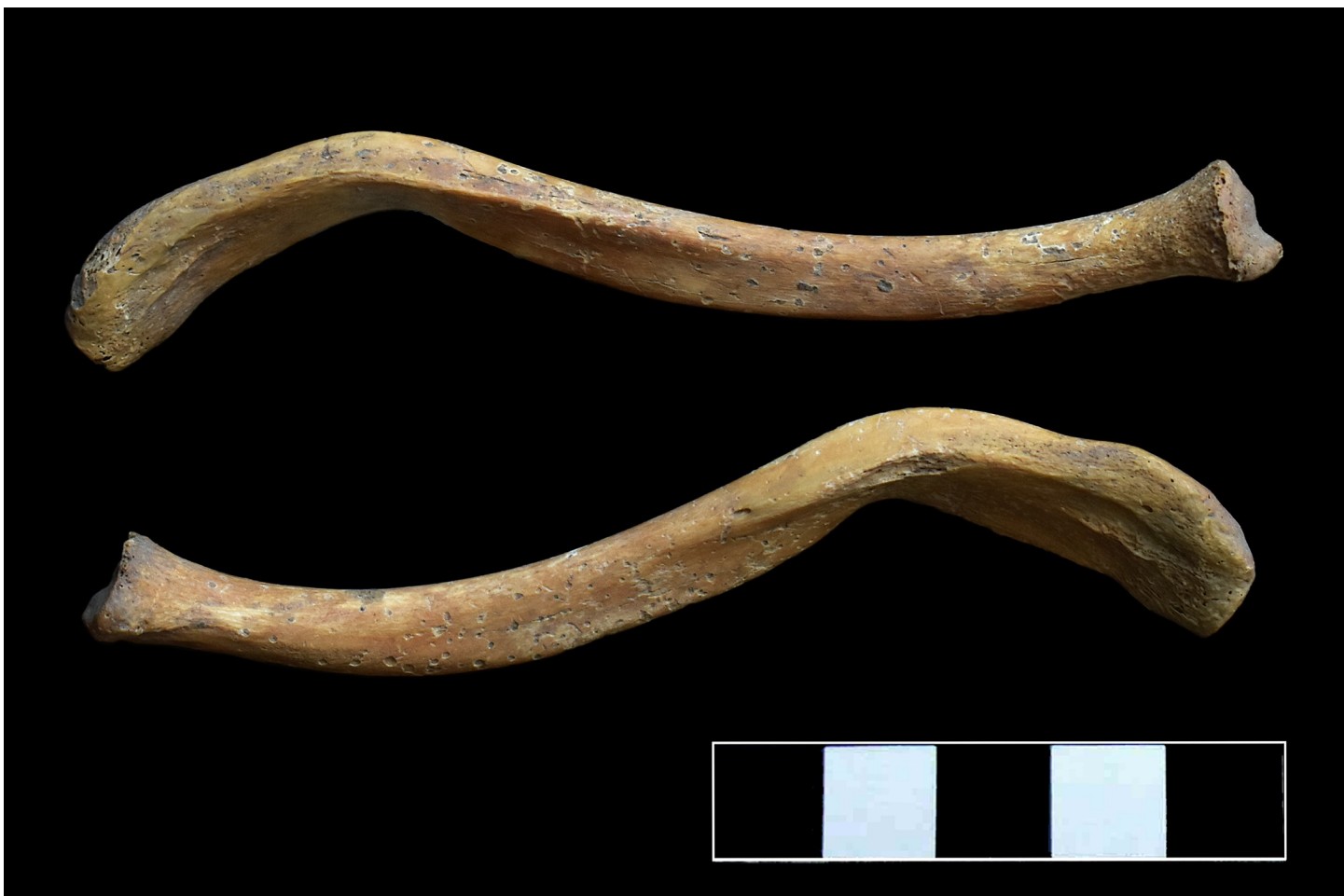

**Fig 11. Abnormal curvature of the clavicles.**

[58, 63, 67]. Although the extra-vertebral cold abscess can remain localized at the initial site of infection, in most cases, it extends vertically (usually downwards) beneath the anterior longitudinal ligament or along the fascial planes [63, 67]. In response to an overlying cold abscess, erosive cortical bone destruction and/or reactive new bone formation can occur on the adjacent bone surfaces (e.g., vertebrae, hip bones, and femora) [63].

The weakening of the affected vertebral bodies due to the formation of osteolytic lesions can result in their consequent collapse under the weight of the trunk [23, 62]. This is characterized by a wedge-shaped appearance, since the cavitation affects predominantly the anterior portion of the vertebral body, with intact or less destroyed posterior vertebral elements [62, 63]. Depending on the spinal location, the collapse of one or more contiguous vertebral bodies may lead to the development of different spinal deformities, most frequently of a sharp angular kyphosis in the thoracic spine (i.e., Pott's gibbus) [57, 60, 63, 67]. The progressive bone destruction and consequent deformity can result in mechanical instability of the spine [59, 60, 66, 67]. To stabilize the vertebral column, subsequent bony fusion of the remnants of the collapsed vertebral bodies and posterior vertebral elements, bony ankylosis of the intervertebral joints, ossification of the intervertebral ligaments, and formation of bony extensions interconnecting the adjacent vertebrae can occur [59, 63].

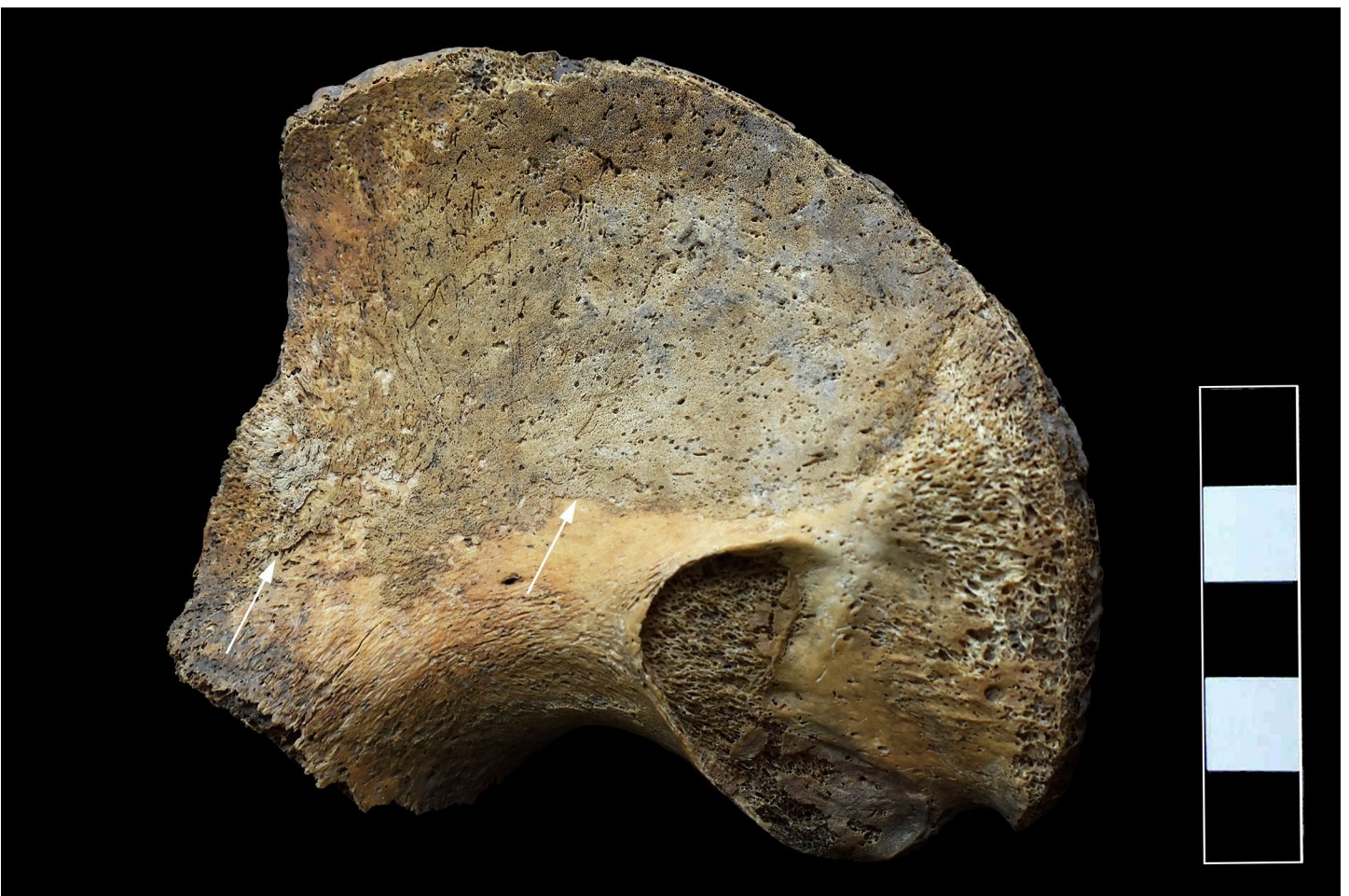

**Fig 12. Reactive new bone formation on the iliac fossa of the right iliac wing (white arrows).**

Although any part of the vertebral column can be affected by TB, the lower thoracic (40–50%) and upper lumbar (35–40%) spine represent the most commonly involved regions in adults [58, 60, 65]. Nonetheless, besides the thoracolumbar junction, the cervicothoracic spine can also become affected quite frequently in children [17, 68, 69]. Spinal TB can be more aggressive and involve more vertebrae in children than in adults; therefore, following vertebral destruction and collapse, they are at particular risk of rapid and severe deformity progression [17, 18, 67, 69, 70]. Moreover, in pediatric spinal TB cases, progression of the deformity even after healing of the disease is not uncommon due to the growing nature of the vertebral column in childhood [31, 57, 65, 67, 71].

In the skeleton of **S0603**, based on the severity and extent of the observed bony changes, the initial site of TB infection could be the upper thoracic spine (T1–5). From this region, the infection could spread (upwards and downwards) beneath the anterior longitudinal ligament–first to the adjacent cervical and thoracic vertebral bodies (C6–7 and T6–7) and later to all true vertebrae (upwards even to the skull base). In later stages of the disease, not only the spine, but the adjacent ribs and soft tissues (e.g., the iliopsoas muscle) could also become affected by direct extension of the infection. TB involvement of the right iliopsoas muscle could result in the development of a paravertebral cold abscess. Later, the TB mass could extend downwards following the course of the right psoas fascia to its tendinous insertion at the lesser trochanter

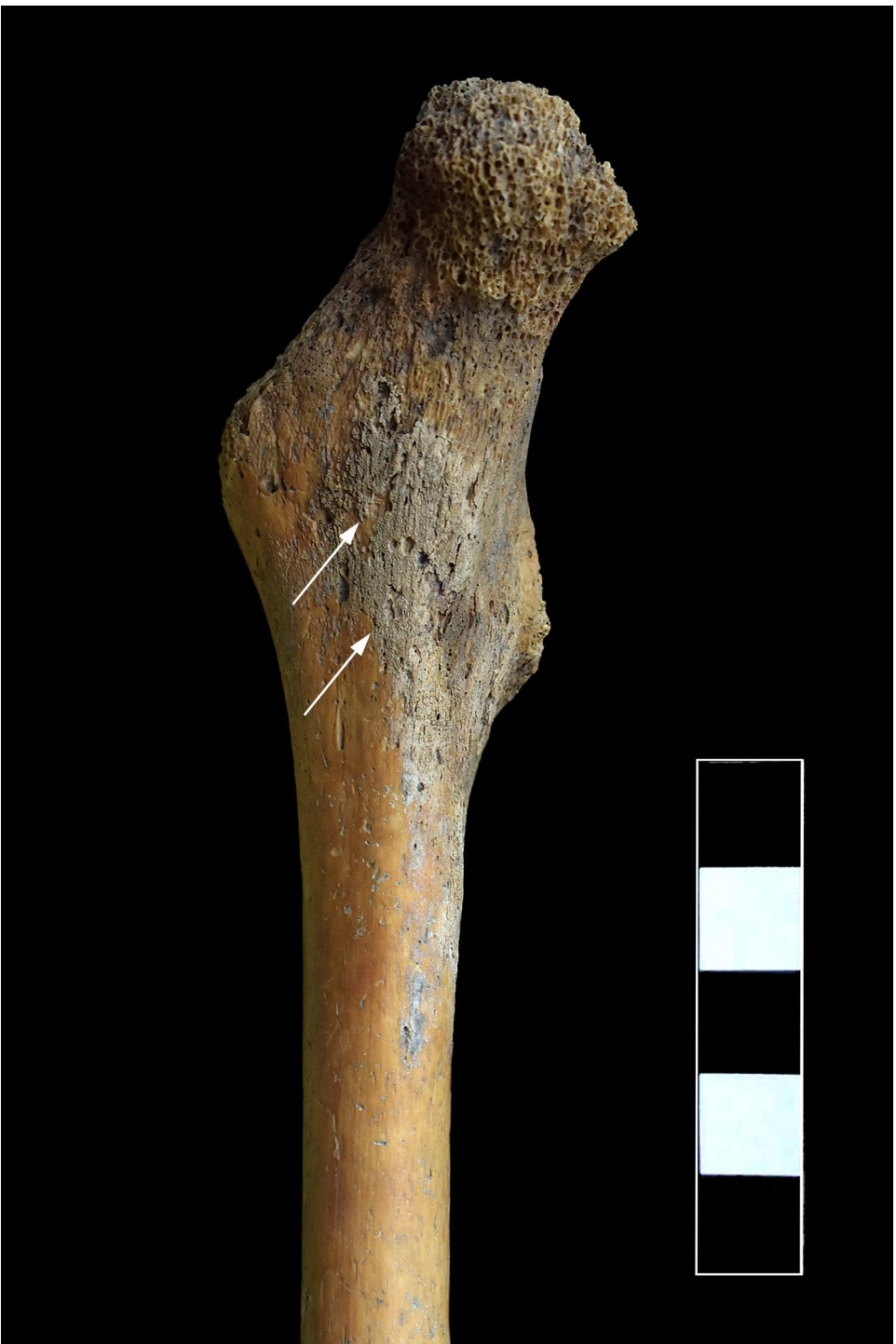

**Fig 13. Reactive new bone formation on the anterior surface of the right femur (proximal end) (white arrows).**

of the right femur. The 12-year-old child's case shows a number of similarities to those of previously described in the paleopathological literature [e.g., 45, 49, 50, 55, 56].

Sparacello and his co-workers [49] reported a probable case of pediatric multifocal OATB (i.e., PO21) from Middle Neolithic Italy. Similarities with **S0603** include involvement of the

spine, ribs, and pelvic area. In the 5-year-old child's vertebral column, the cervical, thoracic, lumbar, and sacral regions were concomitantly affected; however, in contrast to **S0603**, only one or two non-contiguous vertebrae were involved in each region, and the spine did not display angular kyphosis. Similar to **S0603**, bony changes (i.e., pitting, bone resorption with cavitation, and periosteal new bone formation), presumably infectious in nature, were detected in four right-side ribs of PO21; however, they were located not on the vertebral ends but on the sternal ones. Moreover, the shape of the affected ribs was not altered. Although both **S0603** and PO21 exhibited signs of an overlying paravertebral cold abscess (i.e., an iliopsoas abscess) in the pelvic area, in P021, the lesions were bilateral and particularly of a lytic/erosive nature. In contrast to **S0603**, some of the upper limb bones of PO21 (left humerus and right scapula) were also affected by the pathological process.

In a 10-year-old child's skeleton (i.e., L/74) from Roman period Hungary, Hlavenková and her co-workers [55] identified a number of bony changes that are suggestive of OATB. Similar to **S0603**, the most remarkable alterations were discovered in the spine and ribs of L/74. In the 10-year-old child's vertebral column, the sharp angular kyphosis developed in the thoracolumbar region with involvement of five adjacent vertebrae (T9–L1). Due to *post-mortem* damages, no signs of cortical remodeling or reactive new bone formation could be observed on the affected vertebral bodies; nevertheless, ankylosis of the T11–12 intervertebral joints was noted. In contrast to **S0603**, the disease affected only one region of the vertebral column (i.e., thoracolumbar region) and no signs of an overlying paravertebral cold abscess could be detected on the skeletal remains of L/74. Although all the preserved ribs of L/74 presented pathological alterations (i.e., slight or moderate periosteal new bone formation on the visceral surface), their appearance was quite different from those of detected in **S0603**. Only one left-side rib of L/74 showed osteolytic lesions, but not on its vertebral end. In contrast to **S0603**, some of the lower limb bones of L/74 (right femur and both tibiae) exhibited signs of periostitis on the posterior surface of the shaft.

Another probable case of pediatric OATB from Sarmatian period Hungary was described by Marcsik and Kujáni [56]. Similar to **S0603**, the cervicothoracic spine of a 5–6-year-old child (i.e., grave no. 178) was affected by the disease–destruction and collapse of the C4–T2 vertebral bodies and subsequent fusion of their remnants resulted in the formation of a Pott's gibbus. Furthermore, the unequal destruction of the C6–7 vertebral bodies led to scoliosis of the affected spinal region. No other bony changes probably related to TB have been mentioned by the authors.

Matos and his co-workers [45] reported the case of a 12-year-old child (i.e., skeleton no. 8) from medieval Portugal, whose remains exhibited numerous lesions, particularly in the axial skeleton (i.e., spine and ribs), that are consistent with multifocal OATB. Similar to **S0603**, the most severe alterations were observed in the spine. The T3–7 vertebral bodies were completely destroyed, whereas the T2 and T8–10 vertebral bodies displayed flattening, extensive bone resorption, and cavitation. In the T8–L3 region, the posterior vertebral elements were also involved by TB–they demonstrated pitting and osteolytic lesions. Furthermore, ankylosis of some of the intervertebral joints and ossification of the interspinous ligaments between the T7 and T8 vertebrae were noted. The pathological process terminated in a sharp angular kyphosis of the thoracic spine with moderate scoliosis. Similar to **S0603**, disruption of the normal curvature of the 12-year-old child's vertebral column resulted in consequent deformation of the whole thoracic wall–the shape of both clavicles and the 3rd–8th ribs (especially of their vertebral ends) changed. Moreover, direct extension of the infection from the spine to the vertebral end of ribs was also observed in form of destruction, reactive new bone formation, and swelling. In contrast to **S0603**, no signs of an overlying paravertebral cold abscess could be detected on the remains of skeleton no. 8.

In a 9-year-old child's remains from the skeletal reference collection of Lisbon (National Museum of Natural History and Science, Lisbon, Portugal), Gooderham and her co-workers [50] observed bony changes presumably resulted from TB involvement of the skeletal system. Similar to **S0603**, the spine and ribs were most severely affected by the disease. In the T4–12 region, the vertebral bodies exhibited numerous osteolytic lesions that led to considerable bone loss or even complete body destruction (T8–11) with subsequent development of a sharp angular kyphosis and scoliosis. In the same thoracic region, the pathological process extended from the vertebral bodies towards the posterior vertebral elements. Furthermore, the L1–3 vertebral bodies presented swelling. Besides the thoracic and lumbar vertebrae, four right-side ribs (8th–12th) and three left-side ribs (7th, 9th, and 10th) were also affected by TB in form of osteolytic lesions on their visceral surface near the vertebral end. In contrast to **S0603**, no signs of an overlying paravertebral cold abscess could be detected on the skeletal remains of the 9-year-old child. Gooderham and her colleagues [50] noticed that the prolonged disease duration resulted in growth deficit in their case. This could also happen to **S0603**. Therefore, it cannot be excluded that the chronological age of **S0603** was higher than the biological one we could estimate based on the observable skeletal remains.

It should be noted that a number of infectious conditions can lead to the development of vertebral osteolytic lesions; and thus, to the destruction of the affected vertebrae with subsequent formation of an abnormal kyphosis in the spine [58]. In the modern medical practice, it can be very difficult to determine, which particular etiology is indicated by the observed bony changes, due in part to the overlap in manifestations of the aforementioned pathological conditions [41, 45, 49]. Nevertheless, in archaeological cases, it can be even more challenging to establish an appropriate diagnosis, as a lot of diagnostic technologies, which can be used to diagnose living patients (e.g., anamnesis, serological testing, and soft tissue analysis), cannot be applied in the paleopathological practice [45]. Although, based on the modern medical and paleopathological literature, pediatric OATB seems to be the most likely underlying cause of the pathological alterations detected in the skeleton of **S0603**, other infectious etiologies should also be considered in the differential diagnosis. The most relevant ones are granulomatous spinal infections other than TB (fungal infections–e.g., candidiasis, aspergillosis, coccidioidosis, and blastomycosis, and bacterial infections–e.g., actinomycosis and brucellosis) and pyogenic spinal infections [45, 49, 57, 60].

Although in recent years, there has been an increase in incidence of fungal spondylitis, especially in immunocompromised patients, it is still a rare medical condition [72–74]. The most frequent fungal granulomatous infections with potential spinal involvement are candidiasis and aspergillosis that can be found throughout the world [73, 74]. Most cases with *Candida* spondylitis have been reported in adults [75, 76], with the lower thoracic and lumbar spine representing the most commonly affected regions ($\sim$95%); involvement of the cervical spine seems to be uncommon [72–74]. Similar to *Candida* spondylitis, vertebral osteomyelitis due to *Aspergillus* species occurs predominantly in the thoracic and lumbar regions; the disease scarcely affects the cervical spine [72, 74, 77, 78]. Both candidiasis and aspergillosis can cause vertebral body destruction and collapse with subsequent kyphosis formation [62]; and therefore, they cannot be completely rejected as diagnostic options in **S0603**. However, based on their age and localization preference, as well as of their rarity, they seem to be less likely to be responsible for the development of bony changes observed in the spine of **S0603**.

In contrast to candidiasis and aspergillosis, some fungal granulomatous infections with potential spinal involvement are limited to particular geographic areas of the world–among these diseases, coccidioidosis and blastomycosis are the most common ones [72, 74]. Coccidioidosis is endemic to parts of North, Central, and South America [72, 74, 79, 80], whereas blastomycosis is endemic to North America, but there have been increasing reports of the

disease from other parts of the world in the last few decades (e.g., Central and South America, Africa, and Asia) [72, 81–83]. If we assume that about one thousand years ago the geographic distribution of coccidioidosis and blastomycosis was similar to that of today, both diseases can be excluded in the differential diagnosis of **S0603**.

Besides TB, other bacterial granulomatous infections, such as actinomycosis, can cause vertebral osteomyelitis; however, actinomycosis is a rare medical condition, and actinomycotic involvement of the spine is even more uncommon [62, 84–87]. Actinomycotic spondylitis occurs mainly in adults, and is usually secondary to direct extension of adjacent soft tissue infection rather than to hematogenous spread of *Actinomyces* bacteria into the vertebrae [62, 84, 88]. In actinomycotic spondylitis, the cervical and thoracic regions represent the most frequently involved sites, with reactive new bone formation and osteolytic lesions in the affected vertebrae; the intervertebral discs are usually spared [62, 85, 86]. The disease tends to involve the posterior vertebral elements rather than the vertebral bodies–it scarcely results in vertebral body collapse; and thus, consequent kyphosis formation [62, 85]. Based on the above, actinomycosis can be ruled out with high certainty as a diagnostic option in **S0603**.

Similar to actinomycosis, brucellosis is a bacterial granulomatous infection with worldwide distribution [62, 89, 90]. Brucellar spondylitis, most frequently located in the lumbar region, usually affects adults, particularly in their fifth decade of life [62, 90–93]. In brucellar involvement of the spine, destructive and reparative processes occur concurrently, with eventual osteophyte-like reactive new bone formation on the anterior aspect of the affected vertebral bodies [62, 92]. Severe vertebral body destruction and collapse, as well as paravertebral abscess formation are uncommon features in brucellar spondylitis [62, 89, 92, 93]. Therefore, it is unlikely that an infection with *Brucella* spp. resulted in the development of bony changes discovered in the vertebral column of **S0603**.

Pyogenic spinal infection is a common cause of vertebral osteomyelitis, especially in adults over 50 years of age [94–97]. In most cases with pyogenic spondylitis, the lumbar vertebrae are affected, with typical involvement of two contiguous vertebrae and the adjoining intervertebral disc [94–97]. Pyogenic spondylitis usually affects the anterior portion of the vertebral body–destruction of more than half of it, as well as consequent collapse and kyphosis formation are not characteristic features of the disease [95]. Furthermore, in pyogenic spondylitis, involvement of the posterior vertebral elements is relatively rare [97]. Although pyogenic spondylitis cannot be completely excluded in the differential diagnosis of **S0603**, the overall nature and pattern of the observed bony changes indicate that they are more consistent with pediatric OATB.

## Conclusions

Based on the severity and extent of the lesions, as well as on the evidence of secondary healing in the skeleton of **S0603**, the 12-year-old child suffered from TB for a long time prior to death. In **S0603**, OATB resulted in severe body deformation and consequent disability in daily activities, which would have required regular and significant care from others to survive. This implies that in the Árpádian Age community of Győrszentiván-Révhegyi tag, there was a willingness to care for people in need.

From the present-day territory of Hungary, only a few cases with pediatric OATB have been published in the paleopathological literature up to now–one case from the Roman period [55] and two other cases [56] from the Sarmatian period. **S0603** is the first reported case from the Árpádian Age. Since children usually acquire TB from an adult contact with clinically active disease, the presence of a childhood OATB case in the Árpádian Age community of Győrszentiván-Révhegyi tag indicates that other individuals also lived with TB in this historic

population. Moreover, different bony changes probably related to TB infection (e.g., signs of rib periostitis, endocranial alterations, vertebral hypervascularization, and diffuse long bone periostitis) were discovered in a number of other sub-adult and adult skeletons from the Árpádian Age cemetery of Győrszentiván-Révhegyi tag, which also imply that TB was endemic in the community–increased population density and unsanitary living conditions could play a crucial role in the transmission of the disease.

Besides meticulous descriptions from the literature from the first half of the 20[th] century (pre-chemotherapy era), detailed archaeological case studies–like **S0603** –can give us a unique insight into the natural history and different presentations of OATB in children. The establishment of a more reliable and accurate TB diagnosis and the assessment of a more relevant TB frequency in past populations require excessive scientific knowledge on the macromorphological diagnostics of TB. Archaeological case studies, like **S0603**, help us in identifying non-pathognomonic bony changes and/or patterns of lesions that can later be used as macromorphological diagnostic criteria for TB in the paleopathological practice. By providing paleopathologists with a stronger basis for diagnosing TB in ancient human skeletal remains that reveal bony changes resembling that of **S0603** or other published archaeological case studies, a more sensitive means of assessing TB frequency in past populations can be achieved.

## Supporting information

**S1 Video. 3D reconstruction of the severe angular kyphosis (Pott's gibbus) in the T1–6 region of the spine.**
(MP4)

## Author Contributions

**Conceptualization:** Olga Spekker.

**Data curation:** Olga Spekker, Andrea Deák.

**Formal analysis:** Eszter Makai.

**Funding acquisition:** Olga Spekker, György Pálfi.

**Investigation:** Olga Spekker, Andrea Deák, Eszter Makai, Orsolya Anna Váradi, Erika Molnár.

**Project administration:** Olga Spekker.

**Resources:** György Pálfi.

**Supervision:** Erika Molnár.

**Visualization:** Luca Kis, Andrea Deák, Eszter Makai.

**Writing – original draft:** Olga Spekker.

**Writing – review & editing:** Olga Spekker.

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
