## [Decision Letter · Decision Letter 0]

1 Mar 2021

PONE-D-20-32041

An astonishing case of childhood osteoarticular tuberculosis from the Árpádian Age cemetery of Győrszentiván-Révhegyi tag (Győr-Moson-Sopron county, Hungary)

PLOS ONE

Dear Dr.Spakker,

Thank you for submitting your manuscript to PLOS ONE. After careful consideration, we feel that it has merit but does not fully meet PLOS ONE’s publication criteria as it currently stands. Therefore, we invite you to submit a revised version of the manuscript that addresses the points raised during the review process.

We look forward to receiving your revised manuscript.

Kind regards,

Mark Spigelman

Academic Editor

PLOS ONE

Journal Requirements:

Additional Editor Comments:

Dear Dr Spekker

Your paper has been reviewed by 3 highly qualified reviewers all of whom agree it is worthy of publication. however they have suggested some minor changes that will help to tighten the paper please address these and revert back to me.

Prof Mark Spigelman

Reviewers' comments:

Reviewer's Responses to Questions

**Comments to the Author**

1. Is the manuscript technically sound, and do the data support the conclusions?

Reviewer #1: Yes

Reviewer #2: Yes

Reviewer #3: Yes

2. Has the statistical analysis been performed appropriately and rigorously? 

Reviewer #1: N/A

Reviewer #2: N/A

Reviewer #3: N/A

3. Have the authors made all data underlying the findings in their manuscript fully available?

Reviewer #1: Yes

Reviewer #2: Yes

Reviewer #3: Yes

4. Is the manuscript presented in an intelligible fashion and written in standard English?

Reviewer #1: Yes

Reviewer #2: Yes

Reviewer #3: Yes

5. Review Comments to the Author

Reviewer #1: This is a really elegant little case study of a child with probable osteoarticular tuberculosis from medieval Hungary. It is very well written (with a few minor grammatical errors), methodologically sound, and will make a useful contribution to the paleopathological literature on this disease. The detail in which the authors describe the skeletal changes of this disease and the way in which they link the underlying pathophysiology of TB to the specific lesions exhibited by S0603 is exceptionally good – I’d like to see more of this in paleopathological publications! My comments, outlined below, are minor.

Title: This is a personal preference but I would perhaps avoid the term “astonishing”. As the authors acknowledge in the text, there have been other bioarchaeological cases of pediatric OATB with skeletal changes as dramatic as those in S0603.

p. 2, lines 26-27: I’m not sure I would agree that pediatric OATB cases have “scarcely” been described in the bioarchaeological literature and the authors actually spend a considerable amount of time on similar published cases in the discussion.

p. 2, lines 31-33: again, I probably wouldn’t describe the vertebral changes as “remarkable”. The changes exhibited (vertebral collapse secondary to lytic activity) are pretty classic and this is certainly not the only child from the bioarchaeological record who exhibits Potts disease.

p. 2, line 30 “could HAVE/MAY HAVE suffered”

p. 2, lines 44-46: this reasoning seems a bit circular. Paleopathological cases of TB provide a stronger basis for diagnosing paleopathological cases of TB?

p. 4, line 73: minor, but I would add chronological age ranges to “infancy” and “adolescence”.

p. 4, lines 79-80: the primary site of TB infection is always the lungs, except in cases of M. bovis which can begin infection in the gut. I think you mean that it disseminates from another secondary site?

p. 4, lines 90-91: “..early diagnosis IS crucial..”

p. 5, lines 106-111: Again, I think your reasoning is a bit circular here and I’m not sure I’m convinced that paleopathological case studies of very extreme skeletal chances will aid clinicians in making an earlier diagnosis. I understand wanting to make this work clinically relevant but this seems a bit of a stretch.

p. 5, lines 114-116: “..can become more common in the future”. This sentence is a bit awkward. Do you mean that MDRTB is more likely to result in skeletal involvement? If so, do you have a reference for this?

p. 5-6, lines 121-122: is this case really very unique? You describe similar cases of pediatric OATB in the discussion. I think you need stronger justification for why the case you are reporting is different.

p. 7, line 154: I would use the term “skeletal remains” instead of “bone remains” (here and elsewhere in the manuscript.

p. 8, line 181: Define the term “cribra cranii”. Also can you tell whether this is porous new bone, cortical porosity, or trabecular expansion? These distinctions are important for differential diagnosis. The term cribra cranii isn't really used much anymore because it's very non-specific.

p. 9, line 197: I would use the term “trabeculae” or “cancellous bone” instead “spongy material”.

p. 9, line 201: “IN the C6-7 vertebral bodies..”.

p. 10, lines 233-234: “Moreover, they presented swelling.” What does swelling mean? I would describe the changes in terms of the underlying cellular process. E.g. Is this change caused by chronic apposition of subperiosteal new bone?

p. 11, line 251: “..the 12th ribs are post-mortem missing..” this should be “missing post-mortem”

p. 11, line 264: ..”covering the whole pelvic surface”. Pelvic surface doesn’t make much sense anatomically since the whole ilium is part of the pelvis. Maybe interior or medial surface would be better?

p. 12, line 285: “..paravertebral abscess in the pelvic area”. I would tie this back to your lesion descriptions and explain why you think there is a paravertebral abscess (i.e. new bone on the interior ilium and proximal femur). Show your line of reasoning for the reader.

p. 12, line 291: “..Pott’s disease ARISE”

p. 14, lines 323-4: Again, I would avoid “spongy material” and say “trabeculae” or “cancellous bone”.

p. 15, line 362: My understanding is that the term thoracolumbar junction only refers to the articulation between T12 and L1.

p. 21, line 511: “could HAVE sufferED from..”

p. 21, lines 513-516: this information about the case described by Goodman et al and its implications for age estimation probably belongs in the discussion rather than the conclusion.

p. 21, line 517: “..which WOULD HAVE required..”

p. 22, line 518: “THIS implies that in the..”

Figures: I think these would benefit from arrows indicating regions of pathological activity. Otherwise they are very good!

Reviewer #2: The manuscript describes, in detail and at length, an unusual case of childhood tuberculous spondylitis from the Middle Ages. It is of interest but would benefit from more rigour and brevity when describing the pathology. I have annotated the text, suggesting some typographical and other changes.

Reviewer #3: The MS titled "An astonishing case of childhood osteoarticular tuberculosis from the Árpádian Age cemetery of Győrszentiván-Révhegyi tag (Győr-Moson-Sopron county, Hungary)." is a well written, nicely documented MS, showing a rare case of childhood disease. It presents an essential paleopathological case. The method used is proper.

The illustrations help to understand the message. The only suggestion is to change Figure 1/C with a better quality drawing. It is clear that Fig. 1/C shows the original document made during excavation, but a redrawing would it more elegant and improve the quality of the MS. The Hungarian text of the picture ought to be changed into English as well.

In sum, this case study of exhibiting bony changes consistent with osteoarticular tuberculosis is good work and fits well with the aim and scope of the PLOSOne.

Accepting and publishing the article is suggested.

6. PLOS authors have the option to publish the peer review history of their article (what does this mean?). If published, this will include your full peer review and any attached files.

Reviewer #1: No

Reviewer #2: **Yes: **Helen D. Donoghue

Reviewer #3: No

---

## [Author Response · Author response to Decision Letter 0]

6 Mar 2021

Dear Professor Mark Spigelman,

I am very thankful for the reviewers’ insightful and constructive comments regarding our manuscript entitled “An astonishing case of childhood osteoarticular tuberculosis from the Árpádian Age cemetery of Győrszentiván-Révhegyi tag (Győr-Moson-Sopron county, Hungary)” that was submitted to PLOS ONE (manuscript ID: PONE-D-20-32041). I am sure that the reviewers helped us to improve the quality of our manuscript. The main text and several figures have been modified following the reviewers’ suggestions, and the revised files have been uploaded to the submission site of PLOS ONE.

Responses to the suggestions:

1) Reviewer 1 mentioned that in the title, we should avoid the term “astonishing”.” In Reviewer 2’s opinion, we should use the term “unusual” instead of “astonishing”. Following the reviewers’ advice, “astonishing” was changed to “unusual” in the title of our manuscript.

2) Reviewer 1 noted (regarding page 2, lines 26–27) that “I’m not sure I would agree that pediatric OATB cases have “scarcely” been described in the bioarchaeological literature and the authors actually spend a considerable amount of time on similar published cases in the discussion”. We agree with Reviewer 1 that other pediatric OATB cases have also been described in the paleopathological literature, so our case should definitely not be considered as the only one of its kind. However, in our opinion, the number of published pediatric OATB cases, especially in comparison with adult OATB cases, is still low in the paleopathological literature – from Hungary, only three cases have been published up to now. To approximate opinions, we modified the text in the “Abstract” and “Introduction” parts of our manuscript: “not many pediatric OATB cases have been scarcely published in the paleopathological literature”. We hope that Reviewer 1 will be satisfied with the modifications we made.

3) Following Reviewer 2’s suggestion, “i.e.,” was deleted in page 2, line 27.

4) Reviewer 1 commented (regarding page 2, lines 31–33) that we should not describe the vertebral changes observed in our case as “remarkable”. We agree with Reviewer 1 that our case is “certainly not the only child from the bioarchaeological record who exhibits Potts disease”. However, in our opinion, the severity, extent, and pattern of the detected bony changes make our case remarkable – it can be considered as some kind of a “mixture” of the previously published cases. To approximate opinions, “remarkable” was changed to “serious” in the “Abstract” part of our manuscript.

5) Reviewer 1 noted that in page 2, line 30, “could suffer” should be changed to “could have suffered” – unfortunately, we could not find this phrase in the line mentioned above; nevertheless, we think that very likely, we should change “could suffer” to “could have suffered” in page 2, line 39. Reviewer 2 mentioned that our manuscript “would benefit from more rigour and brevity when describing the pathology”. Reviewer 2 suggested that in page 2, line 39, “could suffer” should be changed to “suffered”. Furthermore, Reviewer 2 commented (regarding page 2, line 41) that “could require” should be changed to “would require” and “amount of” should be deleted. To execute the reviewers’ aforementioned suggestions, “could suffer” was changed to “suffered”, “could require” was changed to “would have required”, and “amount of” was deleted. Reviewer 2 also noted (regarding page 2, line 40) that “resulted” should be changed to “results”. In our opinion, modifying our sentence this way would change its meaning – in this sentence, we discussed about our case rather than about OATB cases in general. Therefore, if Reviewer 2 agrees, we would not like to change “resulted” to “results”.

6) Reviewer 1 mentioned (regarding page 44–46) that “… this reasoning seems a bit circular. ”Paleopathological cases of TB provide stronger basis for diagnosing paleopathological cases of TB?”. We agree with Reviewer 1 that in the “Abstract”, our reasoning can seem a bit circular – unfortunately, because of the word limit of the “Abstract” (max. 300 words), we cannot discuss it in more detail in that part of the manuscript. Nevertheless, the “Conclusions” part of our manuscript was supplemented to further highlight this aspect. We hope that Reviewer 1 will be satisfied with this modification and will not find our reasoning circular anymore. 

7) Reviewer 2 suggested that in the first paragraph of the “Introduction”, “to this aspect” (page 3, line 50), “on the one hand” (page 3, line 51), and “on the other hand” (page 3, line 54) should be deleted, and a comma should be inserted after “reactivation” (page 3, line 55) – we modified the text accordingly. Reviewer 2 also mentioned that “on the other hand” (page 3, line 51) should be changed to “however”. In our opinion, modifying our sentence this way would change its meaning – it would imply that there is some kind of contrast between this sentence and the previous one, which is not true. Therefore, if Reviewer 2 agrees, we would not like to insert “however”.

8) Following Reviewer 1’s advice (regarding page 4, line 73), chronological age ranges were added to “infancy” and “adolescence”.

9) Reviewer 1 noted (regarding page 4, lines 79–80) that “the primary site of TB infection is always the lungs, except in cases of M. bovis which can begin infection in the gut. I think you mean that it disseminates from another secondary site?”. We agree with Reviewer 1 that TB primarily affects the lungs, and the hematogenous or lymphogenous dissemination of TB bacteria to other parts of the body, including the skeleton, can result in the development of extra-pulmonary TB forms, such as OATB. However, as we already tried to highlight in the “Introduction” part (page 4, lines 78–83 in the original manuscript), OATB can also result from contiguous spread from adjacent structures (e.g., the infection can spread from the affected meninges to the adjacent cranial bones) or from direct inoculation of TB bacteria into a skeletal site (e.g., as a result of a trauma or a surgical intervention). Therefore, even if only in the minority of OATB cases, besides the lungs, other parts of the body (e.g., the gastrointestinal tract) can be “primary sites” from where TB infection can spread to the bones, or even the bones can represent the primary site of infection. 

10) Reviewer 2 suggested that in page 4, “with” (line 78) should be deleted and “for” (line 96) should be changed to “over”. Furthermore, Reviewer 1 noted that in page 4, “would be” (lines 90–91) should be changed to “is” – we modified the text accordingly.

11) Reviewer 1 mentioned (regarding page 5, lines 106–111) that our reasoning seems to be a bit circular. As we already noted above, to execute Reviewer 1’s comment, the “Conclusions” part of our manuscript was supplemented. Again, we hope that Reviewer 1 will be satisfied with the modifications we made. Reviewer 1 also mentioned that “I’m not sure I’m convinced that paleopathological case studies of very extreme skeletal changes will aid clinicians in making an earlier diagnosis. I understand wanting to make this work clinically relevant but this seems a bit of a stretch.” We agree with Reviewer 1 that the results demonstrated in our current manuscript are more relevant for paleopathologists than for physicians, and that the way we phrased our sentence in the original manuscript may seem a bit of a stretch. Therefore, the sentence was rephrased – we hope that Reviewer 1 will be satisfied with the modifications.

12) Reviewer 2 suggested that in page 5, “could already be” (line 103) should be changed to “were already” and “can” should be changed to “may” (line 116). Moreover, Reviewer 2 noted that in page 5, “evidently,” (line 102), “can” (line 105), “on the one hand” (line 109), and “of” (line 112) should be deleted. Following Reviewer 2’s corrections, we modified the text.

13) Reviewer 1 noted (regarding page 5, lines 114–116) that our “sentence is a bit awkward. Do you mean that MDRTB is more likely to result in skeletal involvement?”. We agree with Reviewer 1 that the way we phrased our sentence can be a little confusing – we did not mean that MDR-TB is more likely to result in skeletal involvement. What we meant is that because the anti-tuberculous drug therapy is not effective enough in a number of MDR-OATB cases, the presentation of the disease can become similar to those of discovered in archaeological cases where no antibiotics could be used to treat TB. Since today MDR-TB becomes more and more common, it can be expected, at least in our opinion, that the “pre-antibiotic-era-like” presentations of OATB will be observed more frequently, as well. To avoid confusion, the sentence was modified – we hope that Reviewer 1 will be satisfied with it.

14) Reviewer 1 asked us if “is this case really very unique?” (pages 5–6, lines 121–122). We agree with Reviewer 1 that in our manuscript, we “describe similar cases of pediatric OATB in the discussion”. Actually, one of the main aims of our study was to compare our case to previously published ones; and thus, highlight its uniqueness. As it was discussed in detail in the “Discussion” part, S0603 shares some similarities with previously published cases – it can be considered as some kind of a “mixture” of them. However, regarding the severity, extent, and pattern of the observed lesions, there are some differences between S0603 and the previously published cases – in our opinion, besides its archaeological context, that is what makes S0603 unique and worth for publishing. We think that not only our case but every archaeological OATB case can give us a unique insight into the natural history and different presentations of the disease (very likely, there are no two cases that are completely identical in every aspect (e.g., severity, extent, and pattern of lesions)).

15) Reviewer 2 suggested that in the “Materials and Methods” part, “extant” should be corrected to “extent” (page 6, line 144), “it is” (page 7, line 149) and “placed” (page 7, line 150) should be deleted, and “is” should be changed to “was” (page 7, line 170). Reviewer 1 mentioned that we should use the term “skeletal remains” instead of “bone remains” in page 7, line 154 and elsewhere in the manuscript (page 7, line 159; page 8, line 177; page 12, line 275; page 17, line 399; page 18, line 440; page 21, lines 515–516; and page 22, line 534). Following the reviewers’ comments, the text was modified. (It should be mentioned that in some sentences (page 7, line 154; page 17, line 413; page 18, line 427; and page 18, line 428), to avoid word repetition, “bone” was deleted before “remains” but “skeletal” was not inserted instead of “bone”.)

16) Reviewer 1 asked us to define the term “cribra cranii”. Furthermore, Reviewer 1 asked us to “tell whether this is porous new bone, cortical porosity, or trabecular expansion?”. Stravopodi et al. (2009) defined cribra cranii “as an osseous disorder manifested as pitting and/or porosis on the cranial vault” – in our manuscript, we used the same definition for cribra cranii. Based on the macromorphological appearance of the lesions observed on the ectocranial surface of the two parietal bones of S0603, they are very likely trabecular expansions. Although in several studies (Masson et al., 2013; Kyselicová et al., 2016), cribra cranii was associated with TB (and a number of other conditions), the exact etiology of cribra cranii is still unknown and should be further investigated (Rivera & Lahr, 2017). In the “Results” part of our manuscript, we wanted to mention all pathological alterations that were noted in the skeleton of S0603; nevertheless, in the “Discussion” part, only those lesions were discussed in detail that can be associated with TB with high certainty. Thus, cribra cranii was left out from the “Discussion” part.

17) Reviewer 1 mentioned (regarding page 9, line 197 and page 14, lines 323–324) that we should use “trabeculae” or “cancellous bone” instead of “spongy material”. Following Reviewer 1’s suggestion, “spongy material” was changed to “trabeculae” (page 9, line 197; page 13, line 305; and page 14, lines 323–324) or “cancellous bone” (page 13, line 295; page 13, line 300; and page 14, lines 331–332) throughout the text.

18) Reviewer 1 noted that “from” should be changed to “in” in page 9, line 201 and “post-mortem missing” should be changed to “missing post-mortem” in page 11, line 251 – we modified the text accordingly.

19) Reviewer 1 asked us (regarding page 10, lines 233–234) “What does swelling mean?”. Reviewer 1 suggested that we should “describe the changes in terms of the underlying cellular process”. In our case, swelling means that besides osteolytic lesions, cortical remodeling, and signs of hypervascularization, the lower thoracic and lumbar vertebral bodies revealed slight expansion (ballooning) – especially supero-inferiorly. Although ballooning of the vertebral bodies was noted in archaeological and modern spinal TB cases (especially in the central type of the disease) (Matos et al., 2011; Esteves et al., 2017), in children, the vertebral endplates often have an outward convex appearance, progressing to concave during growth (Jaremko et al., 2015; Louie et al., 2018). Therefore, it cannot be excluded, that the slight expansion observed in our case was actually a “normal variant” – considering that S0603 suffered from TB for a long time, it cannot be excluded that the disease negatively affected the growth and development of the child’s vertebrae; and thus, the progression of the vertebral endplates from convex to concave was hindered. Although in the “Results” part of our manuscript, we wanted to mention all pathological alterations that were noted in the skeleton of S0603, in the “Discussion” part, only those lesions were discussed in detail that can be associated with TB with high certainty. Thus, swelling of the lower thoracic and lumbar vertebrae was left out from the “Discussion” part.

20) Reviewer 1 mentioned (regarding page 11, line 264) that “pelvic surface doesn’t make much sense anatomically”; and therefore, we should use a better term. Following Reviewer 1’s suggestion, “pelvic surface” was changed to “iliac fossa” in page 11, line 264 and in the figure legend of Figure 12.

21) Reviewer 1 noted (regarding page 12, line 285) that we should tie back the description of the signs of a paravertebral abscess in the pelvic area to our lesion descriptions and explain why we think there is a paravertebral abscess. Following Reviewer 1’s suggestion, we tried to modify the text accordingly – we hope that Reviewer 1 will be satisfied with the modifications we made.

22) Reviewer 1 suggested (regarding page 12, line 291) that “arises” should be changed to “arise”. In our opinion, ‘arises” is correct, since following “spinal TB”, we should use the verb (i.e., arise) in its singular form; therefore, we did not change “arises” to “arise”. We hope that Reviewer 1 will agree with us.

23) Reviewer 1 noted that “My understanding is that the term thoracolumbar junction only refers to the articulation between T12 and L1.”. We agree with Reviewer 1 that in some studies, the thoracolumbar junction is restricted to the T12–L1 region of the spine. However, in other studies its definition is wider – e.g., T10–L2 region (Rajasekaran et al., 2015) or T11–L2 region (Litré et al., 2013). In our manuscript, we considered the wider definition of the thoracolumbar junction. To avoid confusion or misunderstanding, “thoracolumbar junction” was changed to “thoracolumbar region” in page 17, lines 394 and 398.

24) Reviewer 2 mentioned (regarding page 21, line 496) that “Brucella bacteria” should be changed to “Brucella spp.” – we modified the text accordingly. 

25) Reviewer 1 noted that in page 21, line 511, “could suffer” should be changed to “could have suffered”. Reviewer 2 mentioned that our manuscript “would benefit from more rigour and brevity when describing the pathology”. Reviewer 2 suggested that in page 21, line 511, “could suffer” should be changed to “suffered”. To execute the reviewers’ aforementioned suggestions, “could suffer” was changed to “suffered”.

26) Reviewer 1 mentioned (regarding page 21, lines 513–516) that “the information about the case described by Goodman et al and its implications for age estimation probably belongs in the discussion rather than the conclusions”. Following Reviewer 1’s advice, this part was moved to the “Discussion” part and was rephrased and supplemented to fit better in its new place. 

27) Reviewer 1 suggested (regarding page 21, line 517) that “could require” should be changed to “would have required”. The same sentence can be found in the “Abstract” part of our manuscript, where Reviewer 2 noted that “could require” should be changed to “would require”. To execute the reviewers’ aforementioned suggestions and correct the sentence in the same way, “could require” was changed to “would have required” in page 21, line 517. (Moreover, for the same reason, “amount of” was deleted from the sentence in page 21, line 517, as well.)

28) Reviewer 1 suggested (regarding page 22, line 518) that “it” should be changed to “this”. Reviewer 2 noted that “could also live” should be changed to “also lived” (page 22, line 525) and “could be” should be changed to “was” (page 22, line 530). Following the reviewers’ suggestions, we modified the text accordingly.

29) Reviewer 1 mentioned that figures “would benefit from arrows indicating regions of pathological activity”. Following Reviewer 1’s comment, arrows were placed on several figures.

30) Reviewer 3 noted that Figure 1C should be changed to a better quality drawing. Following Reviewer 3’s advice, Figure 1C was redrawn.

31) We have been asked to review our reference list and if we have cited papers that have been retracted, we should remove them and replace them with relevant current references. When we reviewed our reference list, we found one reference (i.e., Jaswani, 2019 – https://journals.indexcopernicus.com/search/article?articleId=2436063) that might have been retracted since we submitted our manuscript in October, 2020. We removed this reference and replaced it with Spiegel et al., 2005.

In the revised version of our manuscript, we tried to execute all suggestions of the reviewers. I hope this new version will be suitable for publication in PLOS ONE.

Thank you again for the reviewers’ insightful and constructive comments and your editorial work!

Sincerely yours,

Dr. Olga Spekker, PhD

corresponding author

---

## [Editor Report · Decision Letter 1]

29 Mar 2021

An unusual case of childhood osteoarticular tuberculosis from the Árpádian Age cemetery of Győrszentiván-Révhegyi tag (Győr-Moson-Sopron county, Hungary)

PONE-D-20-32041R1

Dear Dr. Spekker

We’re pleased to inform you that as you have replied to thee reviewers comments satisfactorily your manuscript has been judged scientifically suitable for publication and will be formally accepted for publication once it meets all outstanding technical requirements.

Kind regards,

Mark Spigelman

Academic Editor

PLOS ONE
---

## [Editor Report · Acceptance letter]

5 Apr 2021

PONE-D-20-32041R1 

An unusual case of childhood osteoarticular tuberculosis from the Árpádian Age cemetery of Győrszentiván-Révhegyi tag (Győr-Moson-Sopron county, Hungary) 

Dear Dr. Spekker:

I'm pleased to inform you that your manuscript has been deemed suitable for publication in PLOS ONE. Congratulations! Your manuscript is now with our production department. 

Kind regards, 

on behalf of

Dr. Mark Spigelman 

Academic Editor

PLOS ONE